# Impacts of future agricultural change on ecosystem service indicators

Sam S. Rabin[1], Peter Alexander[2,3], Roslyn Henry[2], Peter Anthoni[1], Thomas A. M. Pugh[4,5],
Mark Rounsevell[1], and Almut Arneth[1]

[1]Institute of Meteorology and Climate Research / Atmospheric Environmental Research, Karlsruhe Institute of Technology,
Germany
[2]School of Geosciences, University of Edinburgh, UK
[3]Global Academy of Agriculture and Food Security, The Royal (Dick) School of Veterinary Studies, University of Edinburgh,
UK
[4]School of Geography, Earth and Environmental Sciences, University of Birmingham, UK
[5]Birmingham Institute of Forest Research, University of Birmingham, UK

**Correspondence:** Sam S. Rabin (sam.rabin@kit.edu)

**Abstract.** A future of increasing atmospheric carbon dioxide concentrations, changing climate, growing human populations, and shifting socioeconomic conditions means that the global agricultural system will need to adapt in order to feed the world. These changes will affect not only agricultural land, but terrestrial ecosystems in general. Here, we use the coupled land use and vegetation model LandSyMM to quantify future land use change and resulting impacts on ecosystem service indicators relating to carbon sequestration, runoff, biodiversity, and nitrogen pollution. We additionally hold certain variables, such as climate or land use, constant to assess the relative contribution of different drivers to the projected impacts. While indicators of some ecosystem services (e.g., extreme surface water flow levels) see trends that are mostly dominated by the direct effects of climate change, others (e.g., carbon sequestration) depend critically on land use and management. Scenarios in which climate change mitigation is more difficult (Shared Socioeconomic Pathways 3 and 5) have the strongest impacts on ecosystem service indicators, such as a loss of 13–19% of land in biodiversity hotspots and a 28% increase in nitrogen pollution. Evaluating a suite of ecosystem service indicators across scenarios enables the identification of tradeoffs and co-benefits associated with different climate change mitigation and adaptation strategies and socioeconomic developments.

## 1   Introduction

Exploring how the agricultural system might shift under different plausible future climate and socioeconomic changes is critically important for understanding how the future world—with a population increase by 2100 ranging from 1.5 billion to nearly 6 billion people (KC and Lutz, 2017)—will be fed. In addition, land-based mitigation—reducing deforestation, increasing sequestration in natural and agricultural lands, and expanding biofuel use—might be an important piece in the strategy to achieve warming targets laid out in the Paris Agreement (Rogelj et al., 2018; van Vuuren et al., 2018). The implications of resultant shifts in land use areas and management inputs go far beyond food security. Human society depends on a wide range of ecosystem services which broadly fall into three categories (IPBES, 2018a): regulating (e.g., greenhouse gas sequestra-

tion, flood control), material (e.g., food and feed production), and non-material (e.g., learning and inspiration). These have all historically been strongly affected by land use and management.

Declining biodiversity due to the loss and degradation of habitat (Jantz et al., 2015; Newbold et al., 2015) raises moral and ethical questions regarding extinction, represents a loss of a non-material ecosystem service *per se*, and indirectly harms other ecosystem services by impairing system function (Simpson et al., 1996; Tilman et al., 2014; IPBES, 2018b). The conversion of forests and other ecosystems to croplands or pasture has also, by releasing carbon from vegetation and soil pools, caused about a third of humanity's $CO_2$ emissions since 1750 (Ciais et al., 2013). Land-use change also alters how vegetation intercepts rainfall and takes up water from the soil, affecting the amount and timing of runoff and thus water supply and flood risk (Wheater and Evans, 2009; Haddeland et al., 2014). This affects both human and natural systems, as do changes in runoff quality: Nitrogen (N) compounds from fertilizer dissolve in soil water and are transported from agricultural land to freshwater and marine ecosystems. There, this nitrogen pollution can cause eutrophication and affect various ecosystem services, including fishery production (Vitousek et al., 1997). Fertilizer also produces air pollution in the form of nitric oxides (which contribute to respiratory illnesses; Yang and Omaye, 2009) and the greenhouse gas nitrous oxide (the third-largest component of anthropogenic climate change; Fowler et al., 2009; Myhre et al., 2013; Shcherbak et al., 2014). Where nitrogen oxides are elevated, they can react with biogenic volatile organic compounds (BVOCs), which are emitted by plants—especially woody species (Rosenkranz et al., 2014)—for a variety of physiological functions. These reactions produce tropospheric ozone, which is harmful to human health (Ebi and McGregor, 2008), can negatively affect photosynthesis (Ashmore, 2005), and is a greenhouse gas (Myhre et al., 2013). BVOCs also have other, more complicated implications for regulating and material ecosystem services. They can warm the planet by increasing methane lifetime (Young et al., 2009), but on the other hand they help form tropospheric aerosols, which increase reflectance and boost photosynthesis via diffuse radiation (Rap et al., 2018; Sporre et al., 2019). The latter can improve crop yields, but BVOC-enhanced ozone formation can work against that effect (Feng and Kobayashi, 2009).

As global environmental and societal changes continue over the coming decades, it is critical that we understand not just the impacts on the natural world, but how those impacts feed back onto humanity. To explore the possible future evolution of the Earth system and society, models have been developed that simulate the global economy, natural world, and their interactions. A substantial body of research has been built up using such models to examine how future land-use change will affect individual ecosystem services such as carbon sequestration (Brovkin et al., 2013; Lawrence et al., 2018), biodiversity (Jantz et al., 2015; Hof et al., 2018; Di Marco et al., 2019), and water availability and flood risk (Davie et al., 2013; Elliott et al., 2014; Asadieh and Krakauer, 2017). Much less work has been undertaken to evaluate the future of a suite of ecosystem services in an integrated way (Krause et al., 2017; Molotoks et al., 2018). However, such analyses provide critically important evidence for balancing the many competing demands on the land system while achieving climate and societal targets such as those laid out in the Paris Agreement and Sustainable Development goals (Eitelberg et al., 2016; Benton et al., 2018; Verhagen et al., 2018).

Previously, we used the PLUM land-use model to estimate future land use and management change, based on changing socioeconomic conditions as well as climate effects on agricultural yield provided by the vegetation model LPJ-GUESS (Alexander et al., 2018). This coupled model system—the Land System Modular Model, or LandSyMM—is among the state of the art in global land-use change models due to the high level of detail that it considers in the response of agricultural yields to manage-

ment inputs. Whereas most integrated assessment models rely on generic responses of yield to changing climate, atmospheric carbon dioxide, and fertilizer, LPJ-GUESS simulates these processes mechanistically. Land use optimization also happens at a finer grain in LandSyMM (about 3400 gridcell clusters) than in other similar model systems (tens to hundreds of clusters). Finally, LandSyMM is unique in that PLUM allows short-term over- and under-supply of commodities relative to demand (rather than assuming market equilibrium in every year). Here, we take advantage of the mechanistic modeling of terrestrial ecosystems provided by LPJ-GUESS to explore how PLUM-generated future land use and management trajectories—under different scenarios of future socioeconomic development and climate change—differ in their consequences for a range of regulating and material ecosystem services.

## 2   Methods

### 2.1   LPJ-GUESS

The Lund-Potsdam-Jena General Ecosystem Simulator (LPJ-GUESS) is a dynamic global vegetation model that simulates— here, at a spatial resolution of 0.5 degrees—physiological, demographic, and disturbance processes for a variety of plant functional types (PFTs) on natural land (Smith et al., 2001, 2014). Hydrological and most physiological processes are modeled at daily temporal resolution; vegetation growth, establishment, disturbance (including land-use change), and mortality happen annually. Agricultural land is also included, with cropland and pasture being restricted in the types of plants allowed and experiencing annual harvest. Transitions among land use types are given as an input, with LPJ-GUESS calculating the associated change in carbon pools and fluxes (Lindeskog et al., 2013). Four crop functional types (CFTs) are represented: $C_3$ cereals sown in winter, $C_3$ cereals sown in spring, $C_4$ cereals, and rice (Olin et al., 2015a). Nitrogen limitation on plant growth is modeled, with cropland able to receive fertilizer applications (Smith et al., 2014; Olin et al., 2015b). The mechanistic representation of wild plant and crop growth accounts for the $CO_2$ fertilization effect, by which productivity can be enhanced due to improved water use efficiency and (in $C_3$ plants) reduced photorespiration (Smith et al., 2014). In an intercomparison of eight vegetation models over 1981–2000 (Ito et al., 2017), LPJ-GUESS simulated a mean global gross primary productivity very close to the ensemble average, although with the second-steepest increasing trend. LPJ-GUESS has also been shown to realistically simulate the effects of elevated $CO_2$ on temperate cereal yield (Olin et al., 2015b), although the latter effect is stronger than in other crop models (Pugh et al., in prep.). Changes to irrigation, water demand, water supply, and plant water stress as described in the Supplementary Information of Alexander et al. (2018) were included. Most importantly, these changes include (a) increasing maximum irrigation to allow it to bring soil to moisture levels well above the wilting point, and (b) a factor reflecting how soil moisture extraction gets more difficult as the soil gets drier.

LPJ-GUESS simulates variables that can be used as indicators of a number of provisioning and regulating ecosystem services (see also Table 1 in Krause et al. (2017)); these are described in Section 2.5.

## 2.2 PLUM

The Parsimonious Land Use Model (PLUM) is designed to produce trajectories of land use and management based on socioeconomic trends and gridcell-level crop and pasture productivity at a resolution of 0.5 degrees (Engström et al., 2016b; Alexander et al., 2018). Food demand is projected into the future based on external scenario projections of country-level population and gross domestic product (GDP), using the historical relationship of per capita GDP to consumption of each of six crop types—$C_3$ cereals, $C_4$ cereals, rice, oilcrops, pulses, and starchy roots—plus ruminant and monogastric livestock (FAOSTAT, 2018a, b). Demand of a seventh crop type—dedicated bioenergy crops such as *Miscanthus*—is specified based on an exogenous scenario. PLUM calculates the demand for food crops both for human consumption and feed for monogastric livestock, plus any ruminants not raised on pasture.)

Demand is satisfied at the country level by either domestic production or imports, the balance between which is determined considering commodity prices, management costs (fertilizer, irrigation, land conversion, and "other management" such as pesticide use), and changing LPJ-GUESS-simulated productivity due to climate change and $CO_2$ under a range of irrigation-fertilization treatments. The latter are assumed to produce diminishing returns, such that increasing them increases yield at low intensity levels, but less and less so at higher levels, approaching a yield asymptote.

To solve for land use areas and inputs that satisfy demand, PLUM uses least-cost optimization, which allows for short-term resource surpluses and deficits. Such imbalances can be significant in the real world: Global supply of major cereal crops frequently swings 5 to 10% out of equilibrium on an annual aggregate basis, and more extreme imbalances can be seen at the scale of individual countries (FAOSTAT, 2018a). These dynamics are not captured by equilibrium models, such as those used in other land use and integrated assessment models, which represent for each year the stable state that the economic system would move to eventually if the environment did not change. Because global agricultural markets are not in equilibrium, disequilibrium models are needed to capture the real-world process of moving towards—but not reaching—equilibrium in a constantly-changing economic and physical environment. Disequilibrium models have received varying amounts of attention in the literature over time (e.g., Kaldor, 1972; Mitra-Kahn, 2008; Arthur, 2010), and to our knowledge PLUM is the first land use model to incorporate one.

The composition of livestock feed (in terms of which crops are used) is assumed to be flexible, which can result in large interannual fluctuations in demand and production of individual crops as their prices change relative to one another. This is seen, for example, in Supplementary Results Fig. SR7, where oilcrop demand in the US and Canada triples from one year to the next. This assumption is not expected to materially affect the results in terms of gross decadal trends in total agricultural area and management inputs.

As outputs (feeding into LPJ-GUESS for use in LandSyMM), PLUM produces half-degree gridded maps of land use area (cropland, pasture, and non-agricultural land), crop distribution (fraction of cropland planted with each crop type), irrigation intensity, and nitrogen fertilizer application rate. Land use areas are calculated as net change, which neglects certain dynamics—such as shifting cultivation—that can have significant impacts on modeled carbon cycling especially in some re-

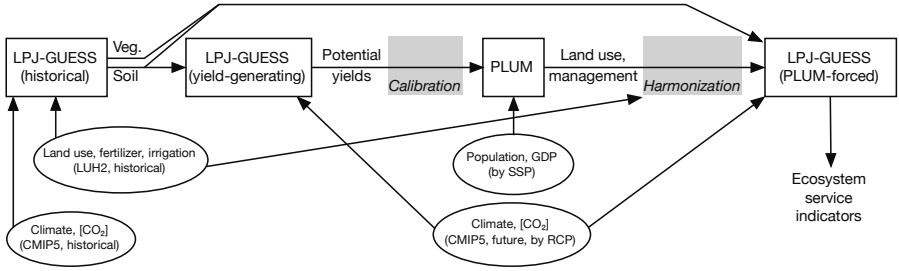

**Figure 1.** LandSyMM structural overview. Ovals represent external input data and white rectangles represent model runs, with arrows indicating data flow from one model run to the next. Gray rectangles represent model coupling processes, whose external inputs have been excluded for simplicity; more information on these can be found in the Supplementary Methods.

gions (Bayer et al., 2017). Other ecosystem services could be affected as well. LandSyMM does not capture these dynamics,
but this was considered an acceptable trade-off for computational efficiency.

## 2.3 LandSyMM: Combining LPJ-GUESS and PLUM

This section and Figure 1 provide an overview of how LPJ-GUESS and PLUM are combined in the LandSyMM runs presented in this work. More details on this coupling can be found in the Supplementary Methods.

The first step in running LandSyMM is to perform "yield-generating" runs in LPJ-GUESS. A simulation of the historical
period generates a model state, which is needed so that vegetation and soil condition can be fed into subsequent runs (Fig. 1). From that state, we perform a series of runs that generate "potential yields" in every gridcell for each crop under six different management treatments in a factorial setup: fertilization of 0, 200, and 1000 kgN ha$^{-1}$ and either no irrigation or maximum irrigation. Changing pasture productivity is accounted for using annual average net primary productivity; for simplicity, we include pasture when using the phrase "potential yields." These potential yields account for changing productivity given changing
climate and atmospheric $CO_2$ concentration.

PLUM then combines the future potential yields from LPJ-GUESS (averaged over five-year timesteps) with its own estimates of future commodity demand to project land use areas, fertilizer application, and irrigation intensity (Fig. 1). PLUM has been found to perform well in this coupled system; its recreation of historical patterns and projections into the future are discussed in Alexander et al. (2018). Here, PLUM's demand estimates are driven by scenario-specific population and GDP data (Sect. 2.4).
The outputs of land use and management from PLUM for a given 2011–2100 scenario are fed into a final LPJ-GUESS run in order to produce projections of the ecosystem service indicators analyzed here (Fig. 1). However, the PLUM outputs must be processed first, because at the beginning of the future period they do not exactly match the land use and management forcings used at the end of the historical period. Feeding the raw PLUM outputs directly into LPJ-GUESS—causing large areas of sudden agricultural abandonment and expansion between 2010 and 2011—would thus complicate interpretation of the results,
especially of carbon cycling. We developed a harmonization routine, based on that published for LUH1 (Hurtt et al., 2011, http://luh.umd.edu/code.shtml), that adjusts the PLUM outputs to ensure a smooth transition from the historical period to the

**Table 1.** Naming convention for LandSyMM runs analyzed in this work, based on land use and management (LU, mgmt.), climate, and $CO_2$ inputs. **Bold** indicates factors held constant in experimental runs. X refers to one of the SSPs (1, 3, 4, 5); YY refers to one of the RCPs (4.5, 6.0, 8.5). Unless otherwise specified, land use forcings are harmonized outputs from PLUM run fed with RCPY.Y-forced LPJ-GUESS potential yields and SSPX socioeconomic data and assumptions, and climate and $CO_2$ forcings are from RCPY.Y.

| Experiment name | LU, mgmt. | Climate | $CO_2$ |
|---|---|---|---|
| sXlum_rYYclico2 (all varying) | 2011–2100 | 2011–2100 | 2011–2100 |
| rYYclico2 (constant LU/mgmt.) | **2010** [a] | 2011–2100 | 2011–2100 |
| sXlum_rYYco2 (constant climate) | 2011–2100 | **1981–2010** [b] | 2011–2100 |
| sXlum_rYYcli (constant $CO_2$) | 2011–2100 | 2011–2100 | **2010** [c] |
| sXlum (constant climate and $CO_2$) | 2011–2100 | **1981–2010** [b] | **2010** [c] |
| rYYco2 (constant LU/mgmt. and climate) | **2010** [a] | **1981–2010** [b] | 2011–2100 |
| rYYcli (constant LU/mgmt. and $CO_2$) | **2010** [a] | 2011–2100 | **2010** [c] |

a: From LUH2 (Hurtt et al., in prep.) and Zhang et al. (2017).

b: Historical (not RCP) climate with temperature detrended. These 30 years are repeated throughout the future period: 2011 uses 1981 climate, 2012 uses 1982 climate, etc.

c: Approximately 389 ppm.

future. While global totals are conserved in almost all cases, harmonization can produce notable differences at the regional scale. Details on this routine can be found in the Supplementary Methods (Sect. SM3).

In addition to the LPJ-GUESS runs forced with harmonized PLUM-output land use and management trajectories, we per-
form several experiments to examine the impact of different factors on the land-use and -management projections generated by PLUM and thus the ecosystem service indicators simulated by LPJ-GUESS in the PLUM-forced runs. By holding either climate, atmospheric $CO_2$, or land use and management constant (or for climate, looping through 30 years of temperature-detrended historical forcings) over 2011–2100, we can estimate the contribution of each to changing ecosystem service indicators in the future. Details regarding the inputs of these experimental runs can be found in Sect. 2.4 and the Supplementary
Methods. The results of these experimental runs were used to inform interpretation of the results but are mostly not presented here. Instead, the Supplementary Results file contains figures (numbers prefixed with SR) supporting claims derived from the experimental runs, in addition to other figures that were not included here to conserve space. Runs are referred to using the naming convention described in Table 1. Note that all PLUM outputs consider LPJ-GUESS yields under changing climate and $CO_2$ concentration, even when those outputs are fed into LPJ-GUESS runs with constant climate and/or $CO_2$. Our analyses
thus account only for the direct effects of changing climate and $CO_2$ on ecosystem service indicators, rather than their indirect effects via land use and management.

## 2.4 Input data and scenarios

The experiments treated here are based around combined future climate-socioeconomic scenarios. Future population growth and economic development are derived from the Shared Socioeconomic Pathways (SSPs; O'Neill et al., 2014; IIASA, 2014).

We use four of the five SSPs, which together cover a wide spectrum of possible storylines for the future evolution of the climate and society (O'Neill et al., 2014, 2015). SSP1 characterizes a world shifting to a more sustainable pathway, with low population growth and strong technological and economic developments. SSP3 describes a pathway with strong population growth and intensive resource usage, low technological development, and lessening globalization. SSP4 is a pathway of inequality with the potential for competition over resources and resource intensification. SSP5 is a pathway dependent on fossil fuels with low

population growth, strong globalization, and high economic and technological growth. (SSP2, a "middle of the road" pathway intermediate between the other four SSPs, is not considered here.)

Scenarios of future climate change and atmospheric $CO_2$ concentrations are based on the Representative Concentration Pathways (RCPs; van Vuuren et al., 2011). SSPs are paired with RCPs based on what sort of climate change could be expected under each SSP's storyline: SSP1 with RCP4.5, SSPs 3 and 4 with RCP6.0, and SSP5 with RCP8.5. RCP numbering refers to

each scenario's average global radiative forcing (W m$^{-2}$) in 2100.

We use climate input data from the Fifth Coupled Model Intercomparison Project (CMIP5; Taylor et al., 2012) outputs of the IPSL-CM5A-MR climate model (Dufresne et al., 2013). Maps of temperature and precipitation change over the simulation period for each RCP are presented in the Supplementary Results (Fig. SR1). The CMIP5 runs did include land-use change, but not the trajectories output by PLUM. As such, and as with all models that are not climate-coupled but rather use offline

forcings, we do not consider the effects of our simulated land-use change on climate. We also use just one climate model, and as such the only uncertainty explored in this work is uncertainty related to scenario choice.

Future socioeconomic data – country-level population and GDP projections – are taken from version 0.93 of the SSP database (IIASA, 2014). Demand of dedicated bioenergy crops such as *Miscanthus* is specified according to the SSP2 scenario from the MESSAGE-GLOBIOM model described by Popp et al. (2017); demand for bioenergy from food crops is specified to double

from 2010 by 2030 and thereafter remain constant. The SSP narratives also affected parameters within PLUM. These included input and transport costs, tariffs, and minimum non-agricultural area (which places an upper limit on the total fraction of a gridcell that PLUM can allocate to cropland and pasture). Values were estimated for each SSP based on an interpretation of the storylines (O'Neill et al., 2015; Engström et al., 2016a) and can be found in Table SM6. Because of these scenario-specific parameters, the raw PLUM outputs are not necessarily expected to match at the beginning of the period.

Historical land use areas (cropland and pasture fractions), irrigation, and synthetic nitrogen fertilizer application levels were taken from the Land Use Harmonization v2 dataset (LUH2; Hurtt et al., in prep.). Historical manure application rates (simplified upon import to LPJ-GUESS as pure nitrogen addition) come from Zhang et al. (2017). Historical crop distributions (i.e., given LUH2 cropland area in a gridcell, what fraction was rice, starchy roots, etc.) came from the MIRCA2000 dataset (Portmann et al., 2010) and were held constant throughout the historical period.

**2.5 Ecosystem service indicators**

LPJ-GUESS simulates a number of output variables that here serve as the basis for quantifying ecosystem services. The carbon sequestration performed by terrestrial ecosystems is measured as the simulated change in total carbon stored in the land system, including both vegetation and soil. Ecosystem nitrogen in LPJ-GUESS is lost in liquid form via leaching (a function

of percolation rate and soil sand fraction), and in gaseous form through denitrification (1% of the soil mineral nitrogen pool per day) and fire. Here we combine these into a value for total N loss. LPJ-GUESS also simulates the emission of isoprene and monoterpenes—the most prevalent BVOCs in the atmosphere (Kesselmeier and Staudt, 1999)—and accounts for three important factors regulating their emission: temperature, $CO_2$ concentration ($[CO_2]$), and changing distribution of woody plant species due to climate and land use change (Arneth et al., 2007b; Schurgers et al., 2009; Hantson et al., 2017).

LPJ-GUESS simulates basic hydrological processes such as evaporation, transpiration, and runoff. The latter is calculated as the amount of water by which soil is oversaturated after precipitation, leaf interception, plant uptake, and evaporation. We present change in average annual runoff as a general indicator of trend in water availability. After Asadieh and Krakauer (2017), we also use the difference between 1971–2000 and 2071–2100 in the 95th percentile of monthly surface runoff ($P95_{month}$) as a proxy for changing flood risk (although note that those authors used daily values), and the difference in the 5th annual percentile ($P5_{year}$) for changing drought risk. Note that we are referring to hydrologic drought, which can be contrasted with, e.g., meteorological drought (a long time with no or little precipitation) or socioeconomic drought (water supply levels too low to satisfy human usage demand; Wilhite and Glantz, 1985). As Asadieh and Krakauer (2017) note, these metrics do not translate directly into impacts due to the mitigation capacity and nonlinear effectiveness of reservoirs, flood control mechanisms, and other infrastructure, as well as changes in demand and mean climate. However, changes in streamflow extremes have served as rough indicators of changing risk in a number of previous global-scale studies (e.g., Tang and Lettenmaier, 2012; Hirabayashi et al., 2013; Dankers et al., 2014; Koirala et al., 2014). While LPJ-GUESS does not model the physical flow of water within and between gridcells, the predecessor LPJ model has been shown to compare well to dedicated hydrological models when aggregated to basin scale (Gerten et al., 2004). As such, where discussing geographic patterns, we will refer to basin-level results only.

Finally, we assess how much land is converted to agriculture within the Conservation International (CI) hotspots, a set of 35 regions covering less than 3% of the Earth's land area but containing half the world's endemic plant species and over 40% of the world's endemic vertebrate animal species (Myers et al., 2000; Mittermeier et al., 2004). These regions each contain at least 1500 endemic vascular plant species and have already lost at least 70% of their original natural vegetation, thus representing highly diverse areas presently at high risk of habitat loss. Note that our chosen metric does not consider areas where agricultural abandonment could lead to a long-term increase in biodiversity, because it is impossible to determine where and how soon, given enough newly available land, there would be sufficient vascular plant richness to qualify as a biodiversity hotspot.

## 3    Results and Discussion

### 3.1    Land use areas and management inputs

LandSyMM simulates net global loss of natural land area over the 21st century in all scenarios (Fig. 2), with SSP3 seeing the greatest loss of area (10%), SSP1 seeing the least (3%), and SSPs 4 and 5 seeing an intermediate loss (6%). These patterns are mostly reversed for pasture area change, in which all scenarios see an increase, although the trajectory for SSP5 is more

similar to that of SSP3. PLUM also simulates net increased cropland area globally in all scenarios, with SSPs 1 and 5 seeing the least increase, SSP4 seeing more, and SSP3 seeing the most.

Cropland expansion happens at a more or less constant rate in SSP3 and SSP4, but these scenarios experience very different trajectories of crop commodity demand: SSP4 approximately levels off around midcentury, whereas SSP3 experiences only a brief slowdown in growth followed by constantly increasing demand through 2100 (Fig. SR2). The majority of the increased demand in the first half of the century is satisfied by fertilizer application, which increases by more than 75% from the 2010s to the 2050s while crop area increases by less than 15%. However, management inputs per hectare in SSP3-60 approximately plateau after midcentury (Fig. SR3), while crop demand rises 16%. Cropland area expands about 10% between 2050 and 2100, with boosted productivity—thanks to climate change and/or $CO_2$ fertilization—helping to satisfy the rest of the increased demand. Since SSP4-60 experiences the same climate and $CO_2$ fertilization but with level crop demand during the second half of the century, management inputs decrease after about 2050. PLUM prescribes lower irrigation rates by the end of the century for most scenarios (Figs. 2, SR3). This is enabled by higher global mean rainfall in all RCP scenarios, as evidenced by the bars for runoff in Figure 4, as well as by improved water-use efficiency for crops other than $C_4$ cereals due to increased $CO_2$ concentrations. Crop demand increase in SSP3-60 outweighs these effects, however, resulting in higher irrigation in that scenario.

Although population growth in SSP5-85 is more than twice that of SSP1-45, PLUM simulates very similar trajectories of global crop demand in both: an increase until about 2040 followed by a decrease for the rest of the century, with SSP5-85 crop demand ending slightly higher. SSP5-85 livestock demand increases about 20% more than in SSP1-45, which explains the rest of the difference in global caloric needs between the two scenarios (Fig. SR2). However, because SSP5-85 experiences much stronger climate change and $CO_2$ increase, the two scenarios differ importantly in how they satisfy their crop demand over the century. Whereas cropland area increases more or less constantly in SSP1-45 (slightly slowing throughout), in SSP5-85 it decreases through about 2050, after which it increases slowly, ending at a slightly lower global extent than in SSP1-45 despite a jump in the early 2090s as feed becomes more important in raising ruminant livestock (Fig. SR4). Crop production remains similar between the two scenarios, especially in the first half of the century, because SSP5-85 applies much more fertilizer and irrigation water per hectare (Fig. SR3). This gap in these inputs narrows in the second half of the century as climate change and the $CO_2$ fertilization effect become even stronger in SSP5-85 relative to SSP1-45, although the latter also begins to increase PLUM's "other management" intensity (representing, e.g., pesticide application).

Figure 3 presents the change in cropland and pasture area over 2010–2100 for each scenario after harmonization. It should be noted that the harmonization process, while preserving global changes in net area change for each land use type, produces more *gross* area change. Where relevant, in this section and the rest of the Results, we will point out where apparent strong regional effects of land-use change result from changes that were not present pre-harmonization.

Several regional patterns in crop area change stand out in Figure 3:

– North America loses cropland in parts of the Great Plains (mainly $C_3$ cereals; Fig. SR5) and the Midwestern U.S. (mainly oilcrops; Fig. SR6) in all scenarios after harmonization. However, this is exaggerated relative to the original PLUM outputs by ca. 1500%, 1700%, 400%, and 800% for SSPs 1, 3, 4, and 5, respectively. Similarly, harmonization inflates

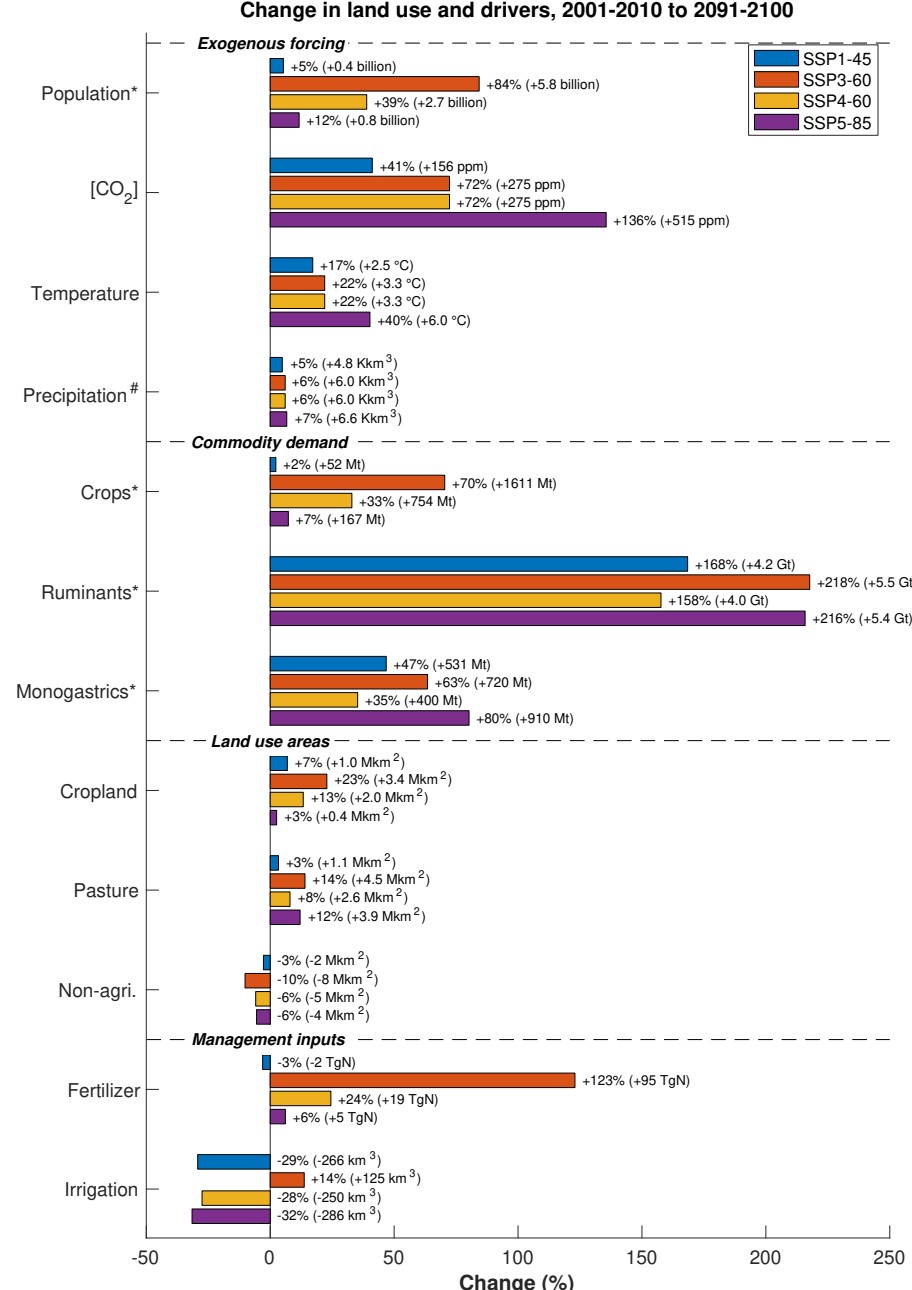

**Figure 2.** Percent change in global socioeconomic, land management, and atmospheric variables between 2001–2010 and 2091–2100. Ruminant demand given in units of feed-equivalent weight. *Asterisk indicates variables whose baseline is 2010 instead of 2001–2010 mean. #The time periods compared for precipitation were 1971–2000 and 2071–2100 due to high interannual variability.

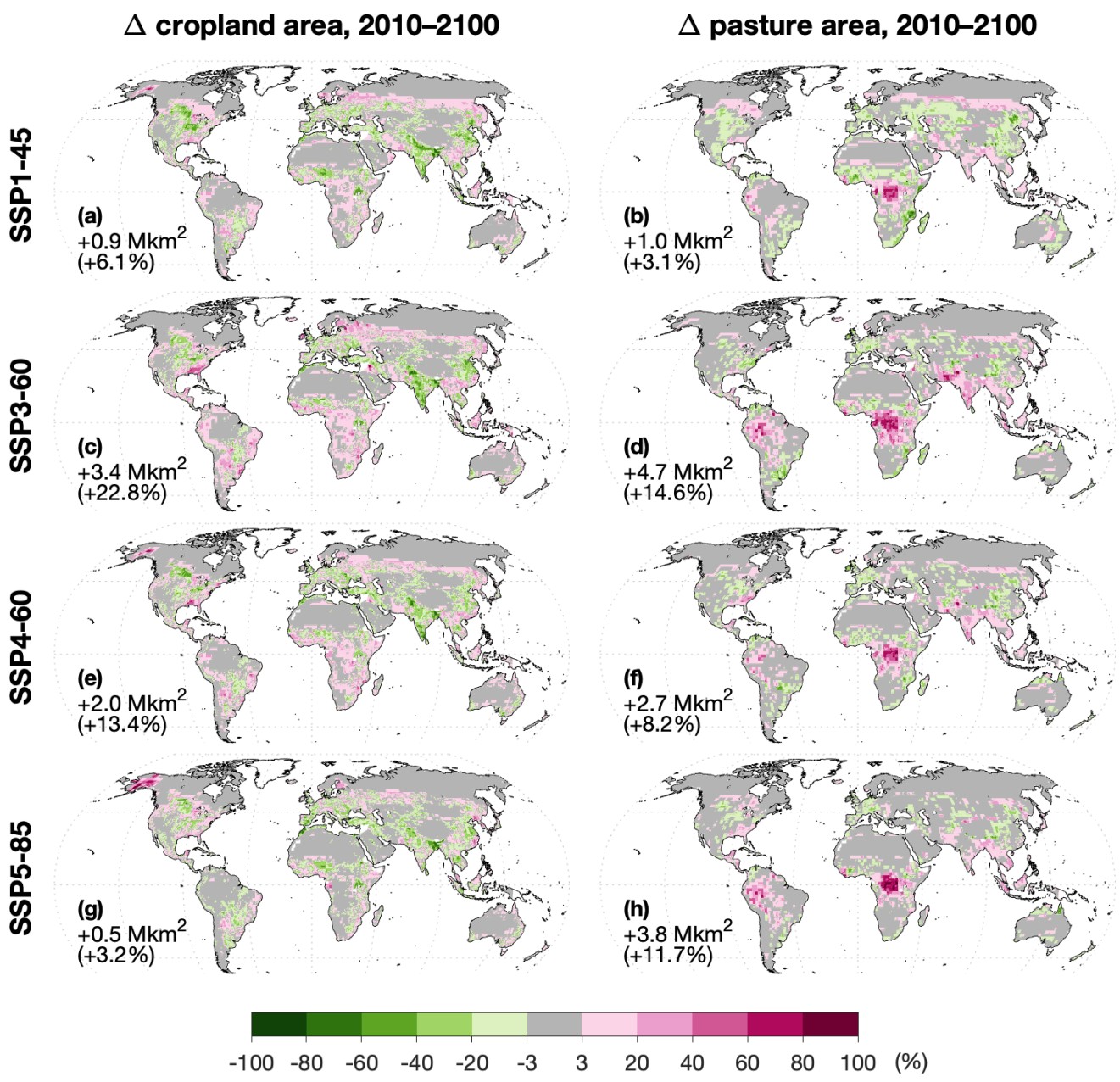

**Figure 3.** (**Left column**) Change in cropland area (as fraction of gridcell) from 2010 (LUH2) to 2100 (harmonized PLUM) under each SSP-RCP scenario. (**Right column**) As left column, but for pasture.

projected cropland expansion in the temperate forests of the eastern U.S. and Canada: by ca. 800% for SSP1-45 and 100% for the other scenarios. On the other hand, large-scale cropland expansion in Alaska in all scenarios except SSP3-60 was almost entirely present in the raw PLUM outputs. This new cropland is entirely planted with spring wheat (Fig. SR5)

and is most extensive in SSP5-85, which sees the largest increase in North American cereal demand—nearly 250% by the end of the century (Fig. SR7)—but also the largest potential yield increase in Alaska, thanks to high warming and $CO_2$ fertilization. Indeed, by the end of the century, the potential yield of rainfed $C_3$ cereals there is similar to or exceeds that of the parts of the Great Plains where cropland is lost (Fig. SR8). It should be noted that while LPJ-GUESS includes several limitations on plant and soil processes based on air and soil temperature, the version used here does not represent permafrost dynamics, and so may be optimistic with regard to the increase in arable land area. However, permafrost extent is expected to decrease across parts of Alaska and the boreal zone as a whole, especially in high-temperature-increase scenarios such as RCP8.5 (Lawrence et al., 2012; Pastick et al., 2015).

– Although crop demand in South Asia (here, India, Sri Lanka, Pakistan, Afghanistan, Bangladesh, Nepal, and Bhutan) increases by more than 100% in SSP5-85 and 170% in SSP3-60 (Fig. SR9), that region net loses large amounts of cropland after harmonization: ca. 30% and 20%, respectively. The raw PLUM outputs saw less loss (8% and 10%, respectively), but the same general pattern. Even so, PLUM projects that the region's crop production would approximately double in both scenarios to satisfy most of the increased demand (Fig. SR9). While some of this is accomplished through increased management inputs in a region where the yield gap is large in the baseline, it also depends markedly on yield boosts due to increased rainfall (Fig. SR1) and rising $CO_2$ concentrations: $C_3$ cereal yields in the constant land-use experiments (`rYYclico2`) triple (RCP6.0) or quadruple (RCP8.5) across large parts of Pakistan and India. This is mostly due to a $CO_2$ fertilization effect, especially in RCP8.5, which sees widespread areas of yield decline when only varying climate (`r85cli`, Fig. SR10).

– PLUM expects sub-Saharan Africa to see crop production increases even larger than South Asia, ranging from +200% in SSP1-45 to +500% in SSP3-60 (Fig. SR11). In contrast to South Asia, nearly the entire region experiences negative yield effects from the changing climate, and the counteracting effect of $CO_2$ fertilization results in yields that are only net slightly boosted in the constant-LU experiments (`rYYclico2`; Fig. SR10). The heightened production comes instead from increased management inputs and, to a much smaller degree, cropland expansion.

– China's crop demand peaks by about 2040; by the end of the century, it has either returned to (SSP3-60) or dropped past 2010 levels (by 30%, 40%, and 25% for SSP1-45, SSP4-60, and SSP5-85, respectively; Fig. SR12). Crop imports decrease from 14% of demand to less than 6%. This fits well with apparent net losses of cropland area in all scenarios, but note that harmonization switched SSP1-45's projection from an 8.5% gain to a 15% loss. Moreover, whereas PLUM projected cropland abandonment to occur in the montane shrublands and steppe of the Tibetan Plateau, after harmonization it occurs throughout the eastern temperate and subtropical forests. Slight cropland expansion projected by PLUM in China's subtropical moist forests is increased 300–600% by harmonization in all scenarios except SSP1-45 (+21%).

Pasture area is projected to expand significantly in the western Amazon in all scenarios (although in SSP1-45 this is strongly exaggerated by harmonization), and even more so in all scenarios in the African rainforest (Fig. 3). This tropical deforestation is largely driven by the increasing consumption of ruminant products in those regions: As incomes increase in developing

tropical countries, PLUM projects greater consumption of commodities such as meat and milk and a reduction in staples such as starchy roots and pulses (Keyzer et al., 2005; Tilman et al., 2011). Depending on the SSP, ruminant products are simulated to account for 23–43% of calories in central Africa by 2100, compared to only 4–7% of calories consumed in 2010 (caloric density derived from FAOSTAT, 2018c). Between 50 and 98% of the ruminant production increase in central Africa goes to this domestic consumption, with the rest being exported.

The African pasture expansion even occurs in SSP1-45, the "Sustainability" scenario (O'Neill et al., 2015), in which LandSyMM simulates a net global pasture expansion of about 1 Mkm$^2$. For comparison, five other land use models all saw SSP1 pasture area decrease: by an average of about 3.4 Mkm$^2$ (Popp et al., 2017). While we do not expect LandSyMM's results to necessarily match those of other models, such a large, qualitative difference requires explanation. Several factors related to experimental setup and overall model structure likely contribute.

First, PLUM makes no assumption about changes in food production needs besides what occurs due to population and GDP changes. The storyline for SSP1, however, with its "low challenges to mitigation," suggests that people will gradually shift to lower-meat diets (O'Neill et al., 2015) than would be expected given GDP levels, at first at least in high-income countries. IMAGE—which simulates a decrease in pasture area of about 7 Mkm$^2$ by the end of the century (Doelman et al., 2018)— incorporates this dietary shift as a 30% (global) reduction in meat consumption relative to what would have otherwise been simulated, and additionally includes a 33% reduction in food supply chain losses to represent efficiencies from improved management and infrastructure (Doelman et al., 2018). Weindl et al. (2017) use the MAgPIE model to show that, under a scenario like ours where historical differences in livestock production efficiency are maintained or exacerbated, a shift to lower-meat diets can reduce the expansion of pasture in sub-Saharan Africa by over 50%.

Second, the land-use modeling components of most integrated assessment models (IAMs)—for example, all those contributing to the LUH2 trajectories (Hurtt et al., 2011)—include demand for timber and other products. The carbon value of forests (and land more generally) can also be included by some, even if forest products are not explicitly modeled (e.g., MAgPIE; Humpenöder et al., 2014), which could come into play in scenarios with policy-based incentives designed to minimize emissions from deforestation and degradation and/or to maximize carbon sequestration. In contrast, PLUM includes neither forest products nor land carbon value. The only cost PLUM considers in converting a forest to agriculture is the cost of conversion, with the opportunity cost of lost forest products or services ignored. Similarly, the only incentive to replace existing agricultural land with forest would be to avoid costs associated with production. Including forest products, payments for carbon sequestration, and managed forestry into LandSyMM could result in more forest simulated over the course of the century. This is especially likely for SSP1, whose storyline specifies a gradual improvement in how the global commons are managed (O'Neill et al., 2015). As an example, IMAGE represented this improvement in SSP1-45 by (a) disallowing clearing of forests with carbon density greater than 200 tons ha$^{-1}$ and (b) reforesting half of the world's degraded or former forest.

The spread in land-use area projections between the most extreme scenarios is much higher in this work than in Alexander et al. (2018), by around 500% for cropland and 700% for pasture. The primary reason for this increase in inter-scenario variation is that Alexander et al. (2018) used the SSP2 socioeconomic scenario for all RCPs, whereas here we compare different SSPs paired with appropriate RCPs. The wide variation among the SSPs in population and economic growth trajectories, along

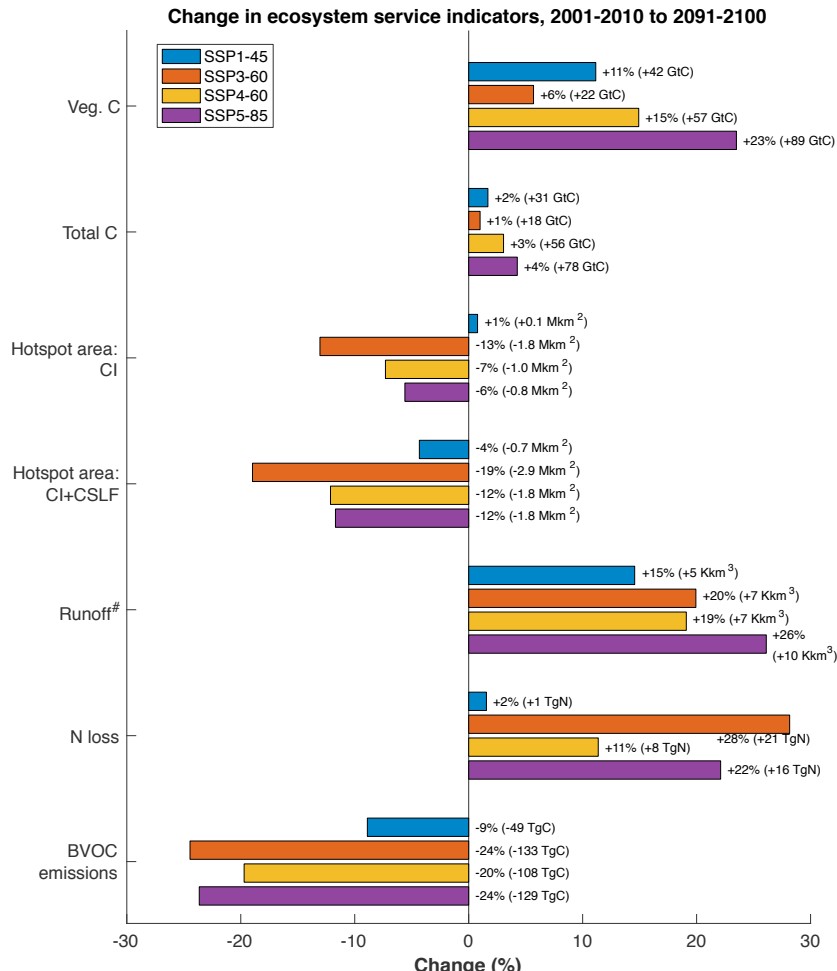

**Figure 4.** Percent global change in ecosystem service indicators between 2001–2010 and 2091–2100. CSLF: Congolian swamp and lowland forests (see Sect. 3.2.5). [#]The time periods compared for runoff were 1971–2000 and 2071–2100 due to high interannual variability.

with SSP-specific PLUM parameters (Sect. 2.4), contribute to this increased spread. Even so, the LandSyMM trajectories are more closely clustered than those from some other land use models. IMAGE, for example, projects a range of cropland area increase from 0.4 Mkm$^2$ to 5.3 Mkm$^2$ across the five SSPs, and pasture trajectories ranging from a decrease of 7.3 Mkm$^2$ to

an increase of 4.4 Mkm$^2$ (Doelman et al., 2018). As described above, IMAGE makes a number of assumptions (based on the SSP storylines) that PLUM does not regarding future deviations from historical "business-as-usual" trends and relationships, including dietary shifts, reductions in food losses during transport, and forest conservation.

### 3.2 Ecosystem service indicators

#### 3.2.1 Carbon storage

Carbon stored in the land system increases for all SSP-RCP scenarios, primarily due to an increase in vegetation carbon (Fig. 4). The increase in each scenario relative to the others depends on both intensity of climate change as well as amount of natural land lost. The large increase of atmospheric $CO_2$ (and thus greater carbon fertilization) in RCP8.5 compared to RCP6.0 means that SSP5-85 has a much greater increase in terrestrial carbon storage than SSP4-60, despite those scenarios having similar trajectories of natural land area (Fig. 2). SSP3-60, which had the most natural land lost but only intermediate carbon 345 fertilization, sees the lowest increase in terrestrial C storage over the century—less than a third that of SSP4-60, which has the same trajectory of changing climate and atmospheric $CO_2$ concentration but a much smaller population increase.

The contrast between effects of changing climate and atmospheric $CO_2$ concentration vs. changing land use and management is starker for vegetation carbon than any other indicator variable examined here. In the constant-LU experiments (`rYYclico2`), vegetation carbon increased 35%, 43%, 43%, and 54% for SSP1-45, SSP3-60, SSP4-60, and SSP5-85, respec-
tively. The experiments with constant-climate and $CO_2$ (`sXlum`), on the other hand, saw respective *decreases* of 5%, 15%, 8%, and 9%.

Vegetation carbon increases are most pronounced in the tropical and boreal forests (Fig. 5) and are due primarily to $CO_2$ fertilization, although increasing temperatures and growing season length also contribute in the boreal zone (Fig. SR13). Extensive conversion to pasture far outweighs any carbon fertilization effect in the African tropical forest, which loses nearly all 355 of its vegetation carbon and up to half its total carbon by 2100 in all scenarios.

Our results for carbon sequestration fall near the lower end of previously-reported projections. Brovkin et al. (2013) examined the change in land carbon storage over 2006–2100 for a number of climate and land surface models. This included IPSL-CM5A-LR: the same IPSL-CM5A Earth system model that produced our forcings, except run at a lower resolution (hence, -LR instead of our -MR). They found that IPSL-CM5A-LR, when forced with emissions and land use change from 360 RCP8.5, simulated uptake of ~400 GtC. This is much greater than our finding of ~89 GtC under SSP5-85 (Fig. 4) despite their land use change scenario (from LUH1; Hurtt et al., 2011) having lost about 30% more non-agricultural land. A rough estimate (not shown) shows that running LPJ-GUESS under RCP8.5 with the same land use change as Brovkin et al. (2013) would have increased total carbon gain by 10–15% at most. Instead, most of the difference is likely because none of the models in Brovkin et al. (2013) limit photosynthesis based on nitrogen availability.

Another study with LPJ-GUESS, Krause et al. (2017), used land use trajectories from the IMAGE and MAgPIE IAMs given RCP2.6 and SSP2, finding an increase in total land carbon pools of 34 GtC and 64 GtC, respectively. Land use scenario played an important role in those results and likely contributes to the discrepancy with ours: Their IMAGE pasture area increased from ~35 Mkm$^2$ to ~40 Mkm$^2$, whereas their MAgPIE pasture area decreased to ~30 Mkm$^2$ and our SSP1-45 pasture stays around ~32 Mkm$^2$. The IMAGE cropland area used in the baseline run of Krause et al. (2017) stayed approximately constant 370 at ~18 Mkm$^2$, as does our SSP1-45's (although at ~15 Mkm$^2$), but their MAgPIE cropland area increased to ~20 Mkm$^2$. Other important differences between the runs in Krause et al. (2017) and ours include our use of different climate forcings and a

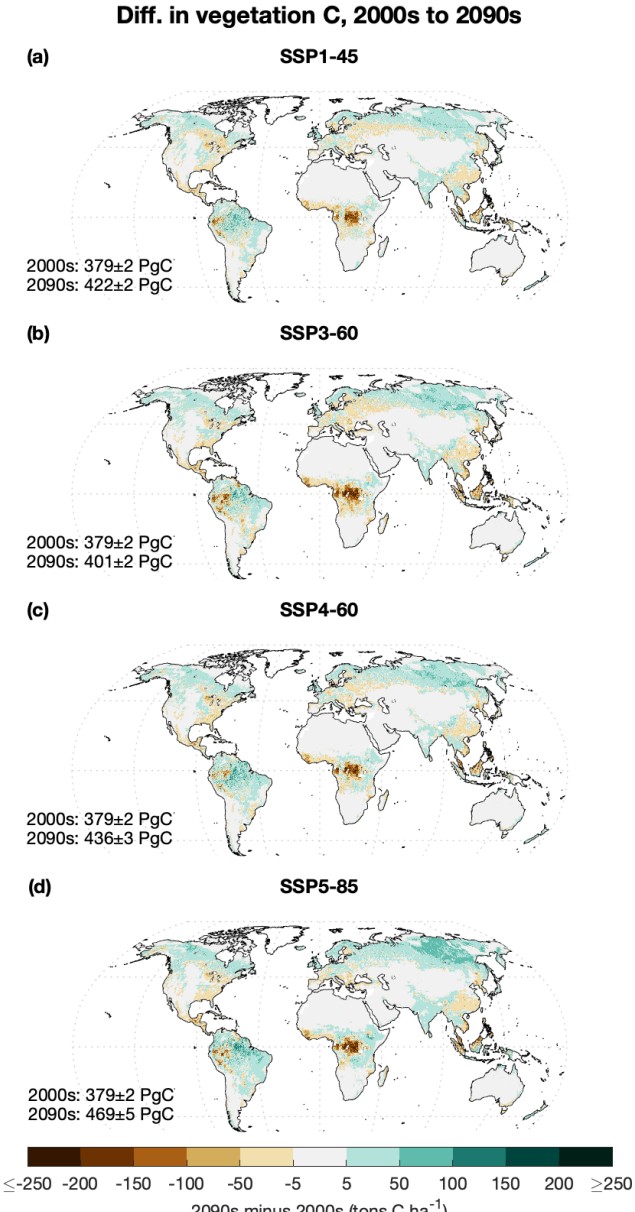

**Diff. in vegetation C, 2000s to 2090s**

**(a)** SSP1-45

2000s: 379±2 PgC
2090s: 422±2 PgC

**(b)** SSP3-60

2000s: 379±2 PgC
2090s: 401±2 PgC

**(c)** SSP4-60

2000s: 379±2 PgC
2090s: 436±3 PgC

**(d)** SSP5-85

2000s: 379±2 PgC
2090s: 469±5 PgC

≤-250 -200 -150 -100 -50 -5 5 50 100 150 200 ≥250
2090s minus 2000s (tons C ha$^{-1}$)

**Figure 5.** Maps showing difference in mean vegetation carbon between 2001–2010 ("2000s") and 2091–2100 ("2090s") for (**a**) SSP1-45, (**b**) SSP3-60, (**c**) SSP4-60, (**d**) SSP5-85. Overlaid text provides decadal means and standard deviations.

different photosynthetic scaling parameter (which accounts for real-world reductions in light use efficiency; Haxeltine and Prentice, 1996).

Krause et al. (2017) used climate forcings from the IPSL-CM5A-LR model, which differs from what we used (IPSL-CM5A-
MR) only in that the former was run at a lower resolution. Both have similar mean global land temperature changes: for RCP8.5,

on the low side of the high end of 18 CMIP5 models examined in Ahlström et al. (2012). This temperature change is strongly correlated with net ecosystem carbon exchange (land-to-atmosphere C flux, excluding fire emissions), so a different choice of climate forcings could have resulted in a stronger C sink or even a C source (Fig. S3 in Ahlström et al., 2012).

### 3.2.2 Runoff

Global precipitation increases in all scenarios (Fig. 2). Again, SSP3 and SSP4 (the two RCP6.0 scenarios) see similar changes; SSP1-45 sees a smaller increase, and SSP5-85 sees the greatest. This pattern is roughly equivalent for changes in global runoff (Fig. 4); comparison of the experimental runs shows that climate change is the most important factor in increasing runoff at a global level in all scenarios (e.g., Fig. SR14). While the impacts of increasing $CO_2$ levels on runoff can be strongly regionally dependent (Zhu et al., 2012), we see that overall more $CO_2$ means less runoff at a global level. A similar result was seen in

two global vegetation models analyzed by Davie et al. (2013), although two others in that comparison showed the opposite effect. Land-use change makes the least difference in terms of global annual runoff, but can be important at the regional level. Deforestation in central Africa, for example, is the primary driver of increasing mean annual runoff there because of reduced evapotranspiration relative to existing vegetation. Note, however, that LandSyMM can only represent the effect of land cover change on evapotranspiration and runoff directly—to include the impact of these flux differences on rainfall would require a

coupling with a climate model.

Such regional patterns in runoff change are arguably more important than global means, since impacts of low water and flooding are actually felt at the level of individual river basins. To evaluate regional impacts, we calculated how much land area was subjected to intensified wet and/or dry extremes (Sect. 2.5). As discussed in Sect. 2.5, these values should not be taken as direct measurements of flooding or drought impacts, but they do serve as useful indicators.

Between 1971–2000 and 2071–2100 under SSP5-85, basins comprising 48% of land area saw increasing flood risk, with a mean $P95_{month}$ increase of 32% (Table 2). Basin-aggregated drought risk increased in 37% of land area, which saw a mean $P5_{year}$ decrease of 58%. At the same time, however, 43% of land area saw decreasing flood risk (mean $P95_{month}$ decrease 42%), and 54% saw decreasing drought risk (mean $P5_{year}$ increase 49%). Other scenarios saw similar fractions of area affected, but smaller mean magnitude of change in flood or drought metric.

Most of the changes in SSP5-85 result from climate change, with some notable exceptions. Land-use change alone contributes notably to increasing drought risk in eastern China, Pakistan, and northwest India (Fig. 6a), although the cropland abandonment driving most of these changes is more densely concentrated pre-harmonization. Agricultural expansion in Alaska and central Africa increase flood risk, while cropland abandonment in southern Pakistan decreases it (Fig. 6b). Similar effects in other regions in Fig. 6—for example, increasing drought risk in Iraq and the central United States, and increasing flood

risk in northeast China—are driven by land-use changes induced mostly by harmonization. (These land-use changes would of course be happening *somewhere*, and thus could still affect runoff similarly, but in a different and potentially more concentrated region.) Land-use change can also serve to counteract the impacts of climate change on runoff. For example, the severity of very low runoff events increases in central America, but it would have increased more if not for the expansion of agriculture there. The effects of land-use change on runoff might be stronger and more widespread if LPJ-GUESS were run coupled with a

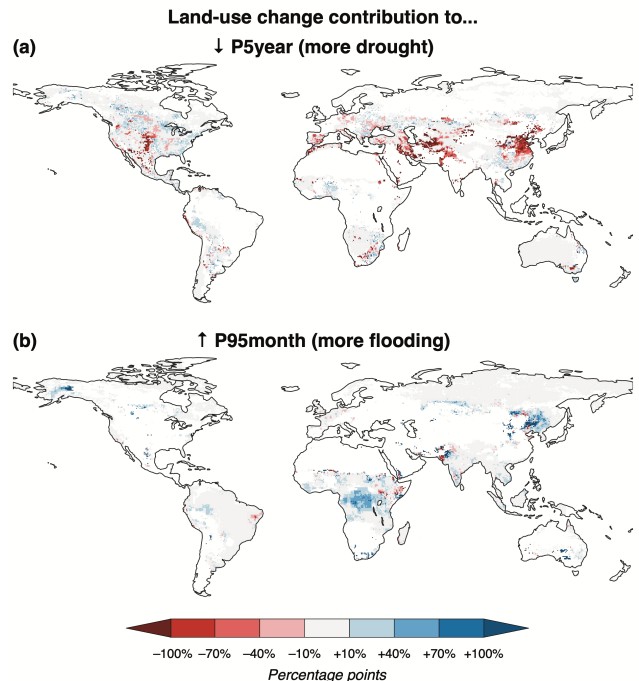

**Land-use change contribution to...**

**(a)** ↓ P5year (more drought)

**(b)** ↑ P95month (more flooding)

−100% −70% −40% −10% +10% +40% +70% +100%
*Percentage points*

**Figure 6.** Contribution of land-use change in SSP5-85 to (**a**) decreasing P5$_{year}$ (drought) and (**b**) increasing P95$_{month}$ (flooding) between 1971–2000 and 2071–2100. White areas either did not have decreasing P5$_{year}$ or increasing P95$_{month}$, respectively, or were excluded due to low baseline runoff (after Asadieh and Krakauer, 2017). Contribution calculated as difference between full run and constant-LU run (i.e., `sXlum_rYYclico2 − rYYclico2`).

climate model, which would account for associated changes in land-atmosphere water and energy fluxes that can have similar impacts on the hydrological cycle as greenhouse gas emissions (Quesada et al., 2017).

Our results for SSP5-85 are compared with the RCP8.5 ensemble from Asadieh and Krakauer (2017) in Table 2. In all categories, LandSyMM finds a mean effect of stronger magnitude. LandSyMM finds less land in basins with increasing drought risk and more with decreasing drought risk than the Asadieh and Krakauer (2017) ensemble. Our results for fraction of land
in basins with increasing or decreasing flood risk are more similar (within 6 percentage points) to those from the Asadieh and Krakauer (2017) ensemble. We expect that our results for land area with increasing and decreasing flood risk would have been lower and higher, respectively, had we used daily values for P95 as Asadieh and Krakauer (2017) did, instead of the LPJ-GUESS-output monthly values. We also expect that the average magnitude of change in those areas would have been closer to zero.

Another important difference between Asadieh and Krakauer (2017) and our analysis is that, whereas that study used five climate models, we used only one. Specifically, compared to 18 other models examined in Ahlström et al. (2012, their Fig. S2), IPSL-CM5A-MR in RCP8.5 simulates a much larger precipitation increase around the Equator, where we see the largest increase in runoff (Fig. SR14a). Finally, LPJ-GUESS is not a full hydrological model: e.g., it does not include river routing.

**Table 2.** Fraction of land with changing drought and/or flood risk between the last three decades of the 20th and 21st centuries in SSP5-85. Numbers in parentheses give each group's mean percent change in runoff. LandSyMM results have been aggregated to basin scale. AK2017: Asadieh and Krakauer (2017).

| Class | LandSyMM | AK2017 |
|---|---|---|
| ↑ drought risk (↓ P5) | 37% (–58%) | 43% (–51%) |
| ↓ drought risk (↑ P5) | 54% (+49%) | 33% (+30%) |
| ↑ flood risk (↑ P95) | 48% (+32%) | 37% (+25%) |
| ↓ flood risk (↓ P95) | 43% (–42%) | 40% (–21%) |

Land surface and hydrological models that include river routing, such as those included in the Asadieh and Krakauer (2017) ensemble, are needed to fully explore how changing precipitation, transpiration, and evaporation actually translate into changes in streamflow and surface water levels.

### 3.2.3 Nitrogen losses

While the evolution of total global nitrogen loss is fairly similar for all scenarios over the first two decades of the simulation, there are notable differences by the end of the century. N loss showed little net change over the century in SSP1-45, whereas SSP3-60 saw a large increase in total N losses. SSP4-60 did not see a notable change in total N losses. SSP5-85 saw an increase in total N losses, although less than that of SSP3-60.

Our N loss at the end of the historical period was similar to that of Krause et al. (2017), but whereas their runs estimated an increase in N losses of 60–80% under RCP2.6, ours under SSP1-45 increased only 2%. Krause et al. (2017) used fertilizer information from IMAGE and MAgPIE . Strong increases in fertilizer in those models resulted in strongly increased yields, but nitrogen limitation is alleviated at much lower levels in LPJ-GUESS. IMAGE and MAgPIE fertilization rates thus often exceeded what plants in LPJ-GUESS could actually take up, resulting in high amounts of N loss. Coupling LPJ-GUESS with PLUM provides for a more internally consistent estimate of future N losses, while still reproducing historical fertilizer application well (Alexander et al., 2018).

One interesting pattern is that climate and management changes can have similar effects on N losses. SSP3-60 sees global fertilizer application more than double by the end of the century, while SSP5-85 fertilizer application at end of the run is slightly lower than in 2011 (Fig. 2). This is reflected in the N losses for the `sXlum` experiments, which increase 25% by the 2090s with SSP3 but only 7% with SSP5. However, in the full runs (`sXlum_rYYclico2`), SSP3-60's N losses increase only about 27% more than SSP5-85's (Fig. 4). This is because the latter experiences higher average global temperatures (increasing gaseous losses) and a greater increase in runoff (increasing dissolved losses), due to the extreme RCP8.5 climate change scenario; in the constant-LU (`rYYclico2`) experiments, N losses with RCP6.0 and RCP8.5 increase by 15% and 24%, respectively. In either

case—but especially under SSP3-60—these increases in fertilizer usage and concomitant nitrogen pollution would exacerbate humanity's already unsustainable impacts on nutrient cycling (Rockström et al., 2009).

### 3.2.4 BVOCs

Global combined BVOC emissions over 2001–2010 totaled ~546 TgC yr$^{-1}$ (~503 and ~43 TgC yr$^{-1}$ for isoprene and monoterpenes, respectively), which compares well with estimates from LPJ-GUESS using different land use scenarios (Arneth et al., 2008; Hantson et al., 2017; Szogs et al., 2017) and the MEGAN model (Sindelarova et al., 2014). Emissions decline in all scenarios: by the most in SSP3-60 and SSP5-85, slightly less in SSP4-60, and the least in SSP1-45 (Fig. 4). This reflects a combination of the effects of land-use change and $[CO_2]$ increases. In the `sXlum` experiments, declines in combined BVOC emissions closely reflect declines in non-agricultural land area (most decrease with SSP3, less with SSP4 and SSP5, and least with SSP1; Fig. SR15). This is a function of the much higher BVOC emissions potential of forests relative to cropland and pasture, as also seen in results from Hantson et al. (2017) and Szogs et al. (2017). In the full runs (`sXlum_rYYclico2`), BVOC emissions decline more in SSP5-85 than in SSP4-60 because the former has higher atmospheric $[CO_2]$, which suppresses BVOC formation (Arneth et al., 2007a). The exact cellular regulatory processes of this "$[CO_2]$ inhibition" remain enigmatic; recent evidence suggests that reduced supply of photosynthetic energy and reductants plays a major role (Rasulov et al., 2016).

Decreases in isoprene emissions are primarily driven by tropical deforestation for agriculture, especially the expansion of pasture in central Africa and South America, and to a lesser extent by the expansion of cropland in the southeastern U.S (Fig. SR16), although the latter is exaggerated by harmonization. The suppressive effect of increasing $[CO_2]$ is mostly counteracted in all RCPs by rising temperatures, which increase BVOC volatility. Monoterpene emissions in what is now tundra increase as woody vegetation expands there, but present-day boreal forests are the main areas of declining monoterpene emissions (Fig. SR17). This is primarily due to the BVOC-suppressing effect of increasing $[CO_2]$, but land-use change also contributes, especially in Alaska.

It is important to keep in mind that the implications of changing BVOC emissions depend on complex, regionally-varying atmospheric chemistry that governs their effects on existing species (e.g., methane) and the formation of secondary products (e.g., ozone and aerosols). The LandSyMM framework, lacking as it does an atmospheric chemistry model, can thus inform only a surface-level discussion of the possible effects of changing BVOCs. However, we wish to provide context for the benefits and detriments associated with changing BVOC emissions, as well as some limitations related to our model setup.

The globally decreased BVOC emissions in all scenarios could contribute a cooling effect in the future, due to expected lower tropospheric ozone concentrations, shorter methane lifetime, and enhanced photosynthesis thanks to more diffuse radiation. This could be counteracted somewhat by warming arising from the reduced formation of secondary aerosols, and it is important to note that the effects on climate are likely to vary from region to region (Rosenkranz et al., 2014). Southeast Asia and the southeastern U.S. are populous areas that could see public health benefits from the deforestation-induced reduction of isoprene emissions and associated ozone levels. However, a sizable portion of that agricultural expansion is for growing bioenergy crops simulated in LPJ-GUESS as *Miscanthus*; BVOC levels would be reduced much less (or perhaps even increased) if woody bioenergy crops were grown instead on the same area (Rosenkranz et al., 2014), but that possibility is not yet included in

LandSyMM. Moreover, the loss of natural land is itself associated with myriad negative health impacts (Myers et al., 2013) which are not simulated in LandSyMM, so it would be shortsighted to view deforestation-induced BVOC reductions as a public health boon. Testing whether and to what extent any of the mechanisms described in this paragraph would make a difference to regional climate and human health would require significant extension of LandSyMM, including the incorporation of new sub-models.

### 3.2.5 Biodiversity hotspots

The large expansion of agricultural land in SSP3-60 has direct consequences for habitat in biodiversity hotspots, which lose over 13% of their non-agricultural land in that scenario (Fig. 4). No other scenario lost more than 8%, and SSP1-45 actually saw a slight gain. However, note that the central African rainforest is not included in the CI hotspots, since that region did not meet the criterion regarding how much of its primary vegetation had been lost (Myers et al., 2000; Mittermeier et al., 2004).

The amount of deforestation projected there in all scenarios—ranging from more than 50% in SSP1-45 to 77% in SSP3-60— could result in great impacts to regional biodiversity. We thus checked how much area is lost if we include the five ecoregions classified by Olson et al. (2001) as Congolian swamp and lowland forests (CSLF), which together roughly correspond to the area of pasture expansion common to all scenarios, into a new "CI+CSLF" hotspot map. This paints a worse picture in all scenarios, nearly doubling hotspot area loss in SSP3-60, more than doubling it in SSP4-60 and SSP5-85, and changing the 1%

gain of SSP1-45 to a 9% loss.

Hof et al. (2018) considered the effects of both climate and land-use change under RCPs 2.6 and 6.0 on species distribution models of amphibians, birds, and mammals. They found that the area of land impacted by these combined threats was approximately equal between the two scenarios for birds and mammals (with more area affected for amphibians under RCP6.0), because although climate change was less detrimental under RCP2.6, to meet such an ambitious climate change target, that

scenario required more land devoted to growing bioenergy crops. We see a similar effect: If ignoring *Miscanthus* area, loss of natural land in CI+CSLF hotspots is reduced (respectively for SSP1-45, SSP3-60, SSP4-60, and SSP5-85) by about 100%, 45%, 39%, and 17%. However, because land cleared for biofuel means less land available for other crops, a full accounting of the contribution of biofuel expansion to land conversion and thus biodiversity would require PLUM runs with no biofuel demand.

It should be noted that area loss in biodiversity hotspots will not necessarily correspond to linear decreases in species richness. Jantz et al. (2015) considered the losses of primary non-agricultural land in the LUH1 land use trajectories (Hurtt et al., 2011), which between 2005 and 2100 were 25% in RCP4.5, 40% in RCP6.0, and 58% in RCP8.5. (Note that Jantz et al. considered only primary land as habitat: i.e., any land that had once been agriculture or experienced wood harvest was "uninhabitable.") However, they translated those values into 0.2–25% of hotspot-endemic species driven to extinction by habitat

loss. This is smaller than the fraction of land area because Jantz et al. (2015) used species-area curves, which model the rate of extinctions per hectare lost as high at the beginning of land clearance in a region but falling as more area is cleared. This nonlinear effect is important to consider, especially considering how much land has (by definition) been cleared already in

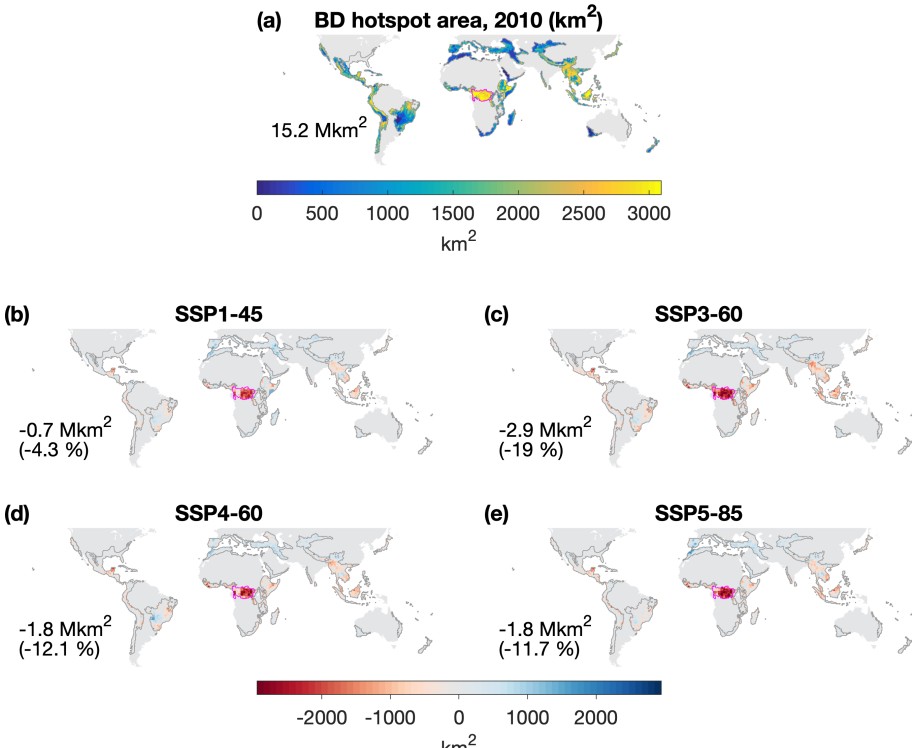

**Figure 7.** (**a**) Area (from LUH2) of non-agricultural land in "CI+CSLF" hotspots in 2010; (**b–e**) change in non-agricultural land area there by 2100 for each scenario. Black outlines indicate CI hotspots; magenta outline indicates CSLF region.

the hotspots, but such an analysis is beyond the scope of the present study. Thus, our numbers for fraction of habitat lost (or gained) should not be considered to translate directly into extinction estimates.

## 4   Conclusions

This work is among the first to comprehensively consider the impacts of future land use and land management change on a suite of ecosystem services under different possible futures of climate and socioeconomic change. Using a uniquely spatially-detailed, process-based coupled model system, we show that scenarios with high socioeconomic challenges to climate change mitigation—SSP3 and SSP5—consistently have some of the most severe consequences for the natural world and the benefits it provides humanity via carbon sequestration, biodiversity, and water regulation. These two scenarios also most strongly affect biogeochemical cycling of nitrogen and BVOCs; while increases in nitrogen losses are generally detrimental, the impact of decreased BVOC emissions is likely to vary regionally. However, various elements of uncertainty—related to PLUM parameter values, global climate model selection, and model design—affect these results and remain to be explored.

Policymakers and other stakeholders need options for how we can meet the needs of a growing and changing society while
achieving climate and sustainable development goals (Benton et al., 2018). Some progress has already been made in this regard
at landscape and global scales (Eitelberg et al., 2016; Verhagen et al., 2018). LandSyMM, and analyses it enables such as the
ones presented here, can be another powerful tool in this aspect of the science-policy interface.

*Code availability.* The code for harmonizing land use and management is available for download on GitHub (Rabin, 2019).

*Author contributions.* All authors contributed to the conceptual design of LandSyMM. S. Rabin composed most of this manuscript, although
all authors contributed to its editing. S. Rabin performed most analyses, with R. Henry helping to interpret PLUM results. P. Alexander
and R. Henry managed PLUM code and performed PLUM runs. S. Rabin made changes as described to LPJ-GUESS code and performed
LPJ-GUESS runs.

*Competing interests.* The authors declare that they have no conflict of interest.

*Acknowledgements.* A.A. and S.R. acknowledge funding from the Helmholtz Association Impulse and Networking fund and the HGF ATMO
programme. P. Alexander and R.H. were supported by the UK's Global Food Security Programme project "Resilience of the UK food system
to Global Shocks" (RUGS, BB/N020707/1). In addition, the authors would like to thank Jonathan Doelman for sharing data about the IMAGE
scenarios (Doelman et al., 2018). This is paper number FILL ON ACCEPTANCE of the Birmingham Institute of Forest Research.

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
