# Peer review of "Impacts of future agricultural change on ecosystem service indicators"

_Earth System Dynamics, 2019_

## Referee Comment (RC1) · Anonymous Referee #1 · 24 Sep 2019

This article presents an ambitious modelling exercise that combines the LPJ-GUESS dynamic global vegetation model with the PLUM land-use model (now under the new name LandSyMM). While the model has been presented before in Alexander et al 2018, the coupled models have not been presented before from the LPJ-GUESS perspective which makes this an interesting article. Additionally, the large number of ecosystem service indicators make the presented analysis interesting on its own.

Before the article can be published in ESD however the structure of the article needs substantial improvement. While I acknowledge the challenge of describing such a complex model in a comprehensive as well as concise manner, I do think improvements can be made. I have three general points and a number of more detailed points that need to be addressed:

- The structure of the methods section is confusing. I would recommend to start with an overview of the models used and how they interact (a diagram might be helpful), then a detailed description of the different ecosystem service indicators analysed. Also, I would recommend to reduce the size of the methods section by moving some of the information on detailed input for LPJ-GUESS and PLUM to the SI. On the other hand, much more information is required on the scenario setup. Details about the main assumptions should be presented. A table with an overview would be helpful. It is important that the reader does not depend on a different article to understand how the presented scenarios are defined.

- The extra experimental scenarios that have been performed to improve understanding of the results are confusing. Maybe these can be added to a descriptive scenario table, either in the main text or the SI, and they could be given explicit names to make it easier to refer to them in the results section, e.g. SSP4-2.6-noCO2/noLUC. I think this is better than for example the long text 'in the constant climate+CO2 experiment' (line 295-296).

- The results and discussion section is rather lengthy and is (as the names correctly states) a mix between results and discussion. I think this reduces the clarity of your story and is not in line with the standard outline of scientific articles. I think the article would greatly benefit from splitting up this section into a clear description of the results (topic by topic) and subsequently a discussion of the results in the context of the literature (again, topic by topic).

Detailed comments:

Line 9-10: this is the first time biodiversity is mentioned in the abstract, while it is presented as one of the major outcomes.

Line 15: please rephrase 'larger than today's by anywhere from' to something more concise and more academic, e.g. 'an increase ranging from 1.5 billion to 6 billion'

Line 57-102: this section describes the ecosystem service indicators presented in the

article. This should be a separate section. In addition, a lot of the text is introduction to why these indicators are important. I think this reduces the clarity of the methods section which should be a more technical description of the indicators presented in the article. Maybe you can move part of the text (in a more concise way) to the introduction.

Line 93-102: I assume the biodiversity indicator is not a standard output of LPJ-GUESS, correct? It is now part of the LPJ-GUESS section which is misleading. It should be more clear that this is calculated based on the downscaled PLUM results.

Line 109-110: The Popp et al 2017 article describes a large number of SSP scenarios from 5 IAM models. Please specify which scenario from which model has been used and preferably refer to a paper that presents the results specifically for this model.

Line 115: this is a very detailed start of this section. Consider restructuring the sentence. Maybe move this entire section to the SI as very detailed information for main text.

Line 137-138: a summary of the climate and land use data used should be given in the text, not in the SI. Consider moving this text to the SI and summarizing input data in the main text.

Line 161: make sure your language is more academic. Don't use terms like 'briefly' or 'attempts'.

Line 178: what is minimum 'non-agricultural area'. This sounds like a PLUM-specific technical term, please rephrase.

Line 203: 'cropland area expands about 10% between 2050,...'. Between 2050 and what?

Line 214: 'what PLUM calls "other management"'. Please use more academic language and avoid usage of model-specific technical terms.

Figure 1: why do you only show ruminants? Non-ruminants also have a very strong

[Figure]

effect on the agricultural system due to high feed requirements. Also, I am surprised about the very strong increase in ruminant demands. FAOSTAT actually shows that in recent decades demand for ruminants-based products has increased relatively little while monogastric-based products have increased much stronger.

Line 243: I was taught to avoid the term 'forecast' (sounds like something a fortune teller would say) but rather use the term 'project', but maybe this is a personal preference.

Line 251-252: this sentence is extremely vague and unhelpful. Please make more explicit.

Line 276: it is impossible to have 500% or 700% higher land use, using current agriculture (~50 Mkm2) this would mean 250 or 350 Mkm2 which is more than the terrestrial area of the world.

Line 282: what do you mean with infrastructure efficiency? Also, if I understand correctly you make similar SSP-specific assumptions in PLUM (SM6). These should also lead to a higher spread in land-use projections, right?

Line 276-278: I don't see why the RCPs lead to a much larger spread in scenario results? Also, is PLUM informed about the yield effects of climate change as these would impact for example trade and food security?

Line 325: is this realistic? Could not reduced feedback effects from lower evapotranspiration from forests in fact reduce runoff?

Line 336-337: how can increased agriculture reduce the risks on droughts? Please explain.

Line 348-349: I do not understand what the 'fraction of included land area' means and why it shows that not including routing is not a big issue (it sounds disconcerting to me). Please explain better.

Line 359-360: why is the estimate more consistent if there is less over application of N? In reality many countries exceed the N uptake rate of plants (most notably China). It sounds less realistic to me that N application only increases by 2%. Does this not imply a major break with historical trends?

Line 369-370: you cannot state 'and other models' here if you refer to three articles that are if I am not mistaken all based on LPJ-GUESS. It is quite logical then that the estimates are similar. Please add an independent reference.

Line 430: I don't think 'storylines' is the right word here as you don't calculate storylines but scenarios that are described by a certain storyline.

Line 435-439: would it be possible to draw stronger conclusions based on the scenario assumptions on how certain future developments should rather be avoided etc?

SI:

Figure SR2: why are the starting points in 2010 so different for irrigation and fertilizer? This should be historic data I assume? Also, there is hardly any trends in the irrigation results, why is this the case?

Figure SR3: I don't understand the top figure on livestock demand. The order of magnitude makes it likely the results are on crop production, but the title suggests this is total production (?) of livestock products. But the bottom figure shows feed for livestock so if the top figure is also about feed demand for livestock it would not be useful. Please explain and improve.

Figure SR8: similar figures are shown for south asia and sub-saharan Africa. Why not consistently show results for all regions? Maybe in slight smaller panels and without the figures with maps in between? Also, why is this figure shown as a delta instead of absolute amounts and how can it be that the demand is so extremely jumpy for oil crops? This seems very unrealistic.

Figure SR10: please write complete description instead of referring to another figure.

[Figure]

---

## Short Comment (SC1) · 2 Dec 2019

Excellent attempt to evaluate scenarios for implications on ecosystem services including trade-off and co-benefits. LandSyMM represents state-of-the-art in modelling of the land-use & vegetation. This type of work is extremely relevant for ecosystem service provision in the context of the Paris Agreement (food-energy-water nexus and beyond). Follows on from earlier studies, e.g. Krause et al., 2017.

Overall I enjoyed reading the manuscript and warrants publication. I found it very informative and represents a substantial amount of work and scientific progress. A minor edit is warranted in the methods section which I found confusing in places. It could be improved as it was difficult to work out exactly how the models link together and

finally the number of runs made. The authors also need to clearly distinguish general text explaining processes from what's actually included in the models. I found the results/discussion section comprehensive and complemented with extremely informative figures. Interesting new advances on attempting to relate flood/drought (see specific comment below), and the link to land loss in biodiversity hotspots.

Minor comments:

28% nitrogen pollution -> aquatic systems, air pollution? Unclear from line 75-90 how nitrogen pollution is quantified/considered in the LPJGUESS (this section is more an introduction to N pollution)

Line 24 "As global environmental and societal changes continue to accelerate over the coming decades,". Somewhat vague. Are we sure changes are and will accelerate. LandSyMM represents state-of-the-art in modelling of the land-use & vegetation. However this advance is not entirely clear. Lines 35-40 could be improved to demonstrate the advance beyond existing IAMs. e.g. "is unique among global land-use change models in the high level of spatial detail that it considers in the response of agricultural yields to management inputs, as well as in allowing short-term over- and under-supply of commodities relative to demand (rather than assuming market equilibrium in every year)" . Vague – what does high-level of spatial detail mean? I wonder if a table contrasting CMIP6 IAMs with LandSyMM would be useful. Also perhaps a real world example can be given to demonstrate the importance of the non-equilibrium assumption (i.e. is this a detail or fundamental). Perhaps the authors can elaborate more on the differences between PLUM and other LU model approaches in section 2.2

Very good coverage of LPJGUESS, however somewhat generic in places. The authors should explicitly state which metrics are calculated for ecosystem services, e.g. how is nitrogen pollution calculated? (at first it gives the impression of some combination of health, water quality, air pollution impacts on vegetation indices; are these processes modelled in LPJGUESS?), i.e. this text gives a more general description of nitrogen

pollution, rather than what is actually modelled in LPJGUESS. Good attempt, but I'm not entirely convinced of the approach for changing flood risk based on monthly runoff – I think this should be raised again in the discussion as a potential limitation/uncertainty. As noted by the authors they apply the Asadieh and Krakauer 2017 approach but use monthly surface runoff data. The authors should state the temporal resolution the model is applied (I assume monthly?). As I understand LPJGUESS outputs per gridcell 1) monthly total N Loss 2) monthly runoff 3) annual land conversion in hotspots, and then interpreted in terms of nitrogen pollution (but no additional metrics used), flood/drought risk (using AK 2017 metric), and loss of land in biodiversity hotspots.

Somewhat confusing section 2.3 on simulation details For example, Line 115-120 "In each scenario, every grid cell is planted with each crop type, each of which is given six different management treatments in a factorial setup: fertilization of 0, 200, and 1000 kgN ha–1 and either no irrigation or maximum irrigation." Why? Somewhat comes out of the blue. Then in 2.3.3 experimental setups Lines 180-190 "In addition to the LPJ-GUESS runs forced with PLUM-output land use and management trajectories (harmonized as described in Sect. 2.3.1), six experimental runs were performed for each scenario, to disentangle the direct effects of climate change (including CO2 concentration increases) from those of land use and management change. " This is another 6 simulations?

Then there's PLUM-forced runs. So I assume this is not the standard approach to running LandSyMM (i.e. PLUM coupled to LPJGUESS), i.e. how frequent is information from LPJGUESS and PLUM exchanged (annually, e.g. potential yields line 167?) Perhaps consider a table with a full list of the simulations would help the reader. Line 140 "The calibration run was forced with climate data from CRU-NCEP version 7 (Le QueÌĄreÌĄ et al., 2016), but with CRU TS3.24 precipitation (Harris et al., 2014) due to problems discovered in the CRU-NCEP precipitation data" What were the problems? (others are using CRUNCEP7 so would gain from this information)

Line 143 IPSL-CM5A-MR – what's the climate sensitivity of the IPSL model? Why

was this selected? (what are the characteristic features of this GCM future prediction), for example which areas are projected to have higher / lower precipitation, as this will govern the simulated flood/drought risk (and affect the other ecosystem services studied?). For example, around line 327 perhaps add some text / speculate on impact of using one climate model for regional runoff. E.g. Figure 5, perhaps it would be useful to include a map of change in precipitation (and temperature) to give the reader a feeling for the importance of choice of GCM.

Line 146 onwards. "Time-evolving historical land use fractions—i.e., the fractions of land in each gridcell that are natural vegetation, cropland, pasture, or barren—were taken from the Land Use Harmonization v2 dataset (LUH2; Hurtt et al., in prep.). The MIRCA2000 dataset (Portmann et al., 2010) provided crop type distributions for the year 2000, which were used for all historical years." I'm a bit confused what is used here. So LUH2 give the cropland coverage, but not which crops, that's given from MIRCA2000 and relative proportion of individual crop types stay fixed through time over the historical period?

Lines 168 onwards describe the SSPs. I think this generic text would work better if it came earlier (SSPs have already been mentioned in several places already).

---

## Referee Comment (RC2) · Anonymous Referee #2 · 20 Dec 2019

The main contribution of this paper is in coupling the PLUM and LPJ-GUESS models to project land-use change impacts in future scenarios, in terms of biodiversity impacts or greenhouse gas emissions.

As such, my main criticism of the paper is that the method section is not very detailed about (1) the assumptions of the two models, (2) the working of the two models, and most crucially (3) how these were combined. While I appreciate the difficulty of communicating complex models in a brief section, seeing that this is the central contribution of the paper, the reader should not be forced to go through the supplemental materials (which is also very densely presented) to understand the models and their interplay. This could potentially be presented as multiple tables and a joint figure exploring the interactions and basic properties of the models.

[Figure]

Conversely, I would suggest to shift large parts of the input data sections to the SM, as (especially in the case of PLUM), these are largely technical details on the modeling side. Instead, the manuscript should spend more time in detailing the scenario setup, as well as how the "holding constant of certain variables" for the purpose of robustness checking was implemented, as based on the abstract and introduction this is a central part of the paper.

Minor comments

- The error bars in Fig. 1 and Fig. 3 are largely cosmetic, as the processes depicted here are highly persistent (e.g. population, cropland), and the error bars merely measure the standard deviations within a decade. The authors themselves do not interpret them within the text, so for the clarity of information they could be also left off. Indeed, if the authors would like to highlight the temporal dynamics, a representation of the whole time-series would be better suited. - In Fig. 2 it would be good to either have a different color scheme for the two columns or the same scale. - In the SM, figures for commodities and exports are presented (SR8, 10,11, 13). Here the trade patterns exhibit highly cyclical behavior, which right not be fully realistic. This should be contrasted with past export dynamics in the same crops and regions.

---

## Short Comment (SC2) · 23 Dec 2019

Review of "Impacts of future agricultural change on ecosystem service indicators" by Sam Rabin et al. The paper presents and explores simulations with the coupled land use and vegetation model LandSyMM to quantify future land use change and resulting impacts on ecosystem service indicators. There is a lot of interesting and thought-provoking material here, and I am sure this paper will create a lot of interest. However, like many "future scenario" papers, there is a lack of consideration of plausibility or uncertainty. The authors do not help the reader to understand why these projections are better or more reliable than other estimates. The section on runoff and flood risk is not convincing, in part because the separation of responses takes no account of what is already known about impacts of $CO_2$ changes on runoff and in part because

no case is made for using mean P95month as a measure of flood risk in a model with no water redistribution instead of relying on a projections using explicit hydrological modelling. The writing style is overblown (particularly in the Introduction) and often obscure (for example in the methods section and in the results section). The messages here could be conveyed in a clearer fashion with some pruning and rewriting, and this would considerably improve the readability of this paper. Shortening the existing text would leave room for a proper discussion section that would allow key issues to be explored. I hope the specific suggestions below can help the authors improve this paper and clarify their arguments, because the reliable estimation of future changes in ecosystem services is important for many purposes and people. Sandy P. Harrison

Specific Comments The Methods section is long, difficult to follow and at the same time does not give sufficient information to allow these experiments to be repeated. I think this needs rewriting, focusing on the information that is really needed to understand what is going on. I think it might be helpful to provide a paragraph at the beginning of this section to explain the logic of the order of presentation - I found some information I expected in one section in somewhere else completely, for example. Some of the information presented could be summarised in the form or a table and/or flowchart diagram, and this would certainly be helpful. Specific comments on individual parts of the Methods are given below. Section 2.1 LPJ-GUESS description. Given how important these simulations are for downstream results, it would be helpful to give a more detailed description of how the model simulates crops (i.e. what are the differences between the treatments of each crop type), how nitrogen limitation is handled, what information is used to specify nitrogen inputs to cropland etc. The information about how irrigation, water demand, water supply, and plant water stress are simulated may well be described in Alexander e al. (2018) but since these are crucial to the current simulations, the approach should be briefly described here. Even the description of how the model simulates natural vegetation types is given short shrift here, so that the claim that it handles $CO_2$ fertilisation is unsupported. It is also unclear from the present description how some of the service "proxies" are calculated by the model. For

example, how does LPJ-GUESS simulate runoff? Please provide a better description of how the model works, so that it is easier to understand its strengths and limitations. Section 2.1 Ecosystem services. Most of Section 2.1 is given over to a description of ecosystem services. What I was expecting here was information about what model outputs were used as indicators of specific ecosystem services. However, much of the text describes why a particular service is important – which should have been information provided in the introduction and indeed partly is provided there. The description of the simulated index is brief and uninformative. What I think would be more helpful would be to reshape this in the form of a table, listing the service and the model output (or outputs). This would save some space which could usefully then be used to provide more details in the model description so that it is clear how these outputs are obtained. Section 2.2. Description of PLUM. Although a detailed explanation of the model is given in Alexander et al. (2018), it would be really nice to know a little more about it here. In particular, I am intrigued about the interface between the two models. What is the handshake, for example, between the four crop types in LPJ-GUESS and the seven crop types in PLUM? This is not explained here, nor is it explained in the description of the simulations. I do not understand how the crop demand optimisation works, and in particular whether this involves considering surpluses and surplus distribution (which should affect commodity prices) or whether it is assumed that there is always a surplus. Section 2.3. Given the complexity of the experimental design, the complicated linking of different models, and the multiple sources of inputs, I think it would be extremely helpful for the reader if you included some kind of flow chart here to guide us through. Section 2.3.3. The factor separation experiments are not well designed. Recycling 30 years of climate is not equivalent to a constant climate. As the results of the FireMIP experiments show, it is difficult to compare these constant climate experiments with constant other experiments when the constant other is based on a single year. Furthermore, the value of treating all climate variables as a single input seems a bit odd when thinking about productivity – it would be more interesting to diagnose what aspects of climate are crucial. In any case, a better factor-separation

approach is needed. Alternatively, given that. these results are "mostly not presented" (line189) you might leave this out.

Results and Discussion section. There is a lot of detail here, but the selection of things to highlight seems somewhat arbitrary. This is particularly the case in the delineation of geographic areas (what, for example, is meant by South Asia?). I was, for example, somewhat surprised by the lack of commentary on changes in China. Given that these kinds of assessments are of largely political interest, I wonder whether there should be some refocussing here - away from biggest changes to most important regions? Some thought should also be given to tabulating results. I would strongly advise separating out the Results from the Discussion, creating a separate section. There are many issues affecting the results presented here, including the impact of methodological uncertainties, that really need to be discussed more fully in this paper. How sensitive are the results to specific inputs? what is the impact of mixing static and time-varying inputs? given that there are large differences between vegetation models in terms of their predictions, how reliable are the LPJ-GUESS productivity estimates? or perhaps, where are they situated with respect to other models? and how much does this matter to the final assessment? How serious is the mismatch between PLUM outputs and the scenarios? How much of an impact does this have on the projections? I am not suggesting that these issues invalidate the study, but I think it would be helpful to discuss the sources of uncertainty and I suggest that you add a Discussion section, where you can do this.

One additional issue that could usefully be included in the Discussion, but certainly needs to be treated somewhere, is the assumption that increased fertilisation will always produce an increase in production rather than a saturating relationship, shown by analyses of field data.

A second issue that could usefully be included is "$CO_2$ fertilisation" – given that this still appears to be controversial, that there is confusion about this is photosynthesis or WUE, that different models produce different strengths of fertilisation and so on.

[Figure]

In comparing LandSyMM results with other models, it would be useful to include a discussion of the. plausibility (or otherwise) of their/your assumptions. This would also deal with the questions: given that there are other simulation results, what does this paper add? and why should we believe the results are more plausible?

I would seriously consider taking out the section 3.2.2, but in any case it needs rewriting. Runoff. The impact of $CO_2$ on runoff is going to be strongly dependent on whether we are talking about semi-arid regions or not, and there is now considerable literature on this (which should be cited). I think a more logical way to organise this section would be around climate regions. The transition from global runoff increasing to "flood" and "drought" risk is abrupt and it would be helpful to actually explain regional patterns of runoff change first. The fact that LPJ-GUESS is not a proper hydrological model, i.e. it does not transfer water between grid cells, it does include groundwater recharge, it does not include surface storage etc. etc. is mentioned in passing here (line 345). But this is a key issue about what "runoff" means and what "flood risk" means. This has been alluded to earlier on by referring to meteorological flood/drought, but it potentially very mis-leading – not for the immediate readers of the paper but certainly for the "assessments" that will pick these results up and re-use them. The logic of focusing on biodiversity hotspots is different from the logic employed with other ecosystem services, in the sense that with the other services you allow for increases/decreases and for changes in geographic regions where increases/decreases can happen. Wouldn't this be a useful approach here too? Is it possible that there would be increases in biodiversity in some regions that are not currently considered hotspots?

Conclusion. If you split the. results and discussion section into two, the you could consider including the conclusions in your discussion section. The current conclusions are not very startling (storylines with high socioeconomic challenges to climate change mitigation consistently have the most severe consequences for ecosystem services) or are simply a repeat of how important this information could be (which was already in the introduction).

Minor comments

Line 15-16. The statement about future population changes is expressed rather badly and is difficult to grasp, please rephrase. Line 25. Is this really a feedback sensu stricto? Lines 47-48. The processes operate on the plant functional types rather than among them. Can you rephrase this to describe the model more clearly. Line 51. When you say C3 cereals sown in winter and spring, presumably these are considered as two PFTs, so it would be clearer to say " C3 cereals sown in winter, C3 cereals sown in spring ...." Lines 68-75. It is impossible to judge whether these measures provide reasonable proxies for water availability, freshwater ecosystem condition, or flood risk because there is no information on how runoff is generated in LPJ-GUESS. is runoff simply the difference between P and ET in a gridcell? or is there transfer of surface water between gridcells? is there a contribution from groundwater? Line 75. If you mean that hydrologic drought is not the same as meteorologic or socioeconomic drought, why not simply say so? This sentence is unnecessary, and begs the question: what is e.g. socioeconomic drought. Line 76-81. How does LPJ-GUESS calculate total nitrogen loss? Do you separate out nitrogen loss from natural ecosystems and agricultural systems? Line 82. The linking of climate change and human health here led me to believe that you were going to look at ecosystem services that mitigated the impact of climate change on human health. Apart from the mention that BVOCs affect ozone which in turn can have impacts on health, you don't really go into this in any depth. For example, you don't mention e.g. mineral dust and the role that vegetation plays on mitigating dust emission pace China. Perhaps changing the emphasis here to plant emissions (which have multiple effects, including on climate and on health) would be a better way to introduce this section. Line 94-102. I can see why the focus on the hotspots is attractive, but in this modelling framework is would also be possible to make a more general assessment of biodiversity loss and this would also be valuable. Line 109. The plant name Miscanthus should be in italics.

Line 115. Is 500 years really sufficient to bring the carbon pools into equilibrium? or

is the phrase a realistic starting point mean to imply that they are not necessarily in equilibrium?

Line 125-126. This first sentence should be moved into the description of the model, as confirmation that PLUM works reasonably well. It is not relevant to a description of the modelling protocol.

Line 127. Given that PLUM has 7 crop types and LPJ-GUESS only four, how do you input PLUM land use into LPJ-GUESS?

Line 129. Please can you bring this flowchart and the table into the main text?

Lines 133-135. Please indicate that the details for these sensitivity tests are given in a following section and reference the section here.

Line 139. Surely this should be: Viovy, N. 2018. CRUNCEP Version 7 - Atmospheric Forcing Data for the Community Land Model. Research Data Archive at the National Center for Atmospheric Research, Computational and Information Systems Laboratory. http://rda.ucar.edu/datasets/ds314.3/.

Line 140. Either spell out what these problems are or refer to a paper that does. Maybe Tang et al. (2017)?

Line 147-149. I am having difficulty with this description. You use time-varying allocations of cropland area per gridcell but a static data set of what these crops were. How did you apply this? Simply assuming that the area might change but the crop remains the same? How much uncertainty does this introduce in your results? Unless you address this in a Discussion section (as suggested above) you need to say something here.

Line 148. Is this still in prep.?

Line 154-155. This sentence is a bit unclear. The manure N is held constant in the calibration run but varying in the other runs?

Line 179-180. Did you estimate these values or are they provided?

Line 184-185. Recycling 30-years of climate is not "constant climate"

Line 210. "first two or so ...." please state what period it actually decreases over.

Figure 2. This is unreadable at the size reproduced here and, with the grey background, the paler colours are not sufficiently visible. You need to find a way of making the changes more visible. Maybe splitting this into two figures would help. (Note the same comments apply to Figure 4, 5).

Line 222. I confess that I find this agricultural expansion in Alaska a bit implausible, even for a high-end scenario, given the topographic constraints and the issue of permafrost. I would be intrigued to know when the permafrost disappears in this scenario. And how infrastructure (or lack of it) would impact this expansion?

Line 228. Since South Asia is not a widely-recognised geographical term, it would be helpful to define where exactly you mean here. Are you including southern China here?

Line 234. What climate change produces more favourable growing conditions in South Asia?

Line 236. Even larger ... even larger than what?

Line 250 et seq. I find this discussion of other model results here confusing. I think you want to separate this from the presentation of all the results from your experiments and move this type of comparison into a separate discussion session. This would allow you to discuss the plausibility of the other assumptions compared to the assumptions encapsulated in your simulations.

Line 267. I do not understand what you mean by "friction" here.

Line 269-274. Please take out this speculation about the impacts of including forest products in LandSyMM based on work that has not yet been done.

[Figure]

Line 275-283. And so what? You appear to be saying that you have different results from one study because they used unrealistic inputs, and that you have different results from another study because they made a set of different assumptions. In the first case, perhaps you could assume that a reader might guess that your results are "better", although you never actually use the "unrealistic" word. In the second case, however, you might give use a hint about the assumptions made by IMAGE would produce more/less realistic results and why.

Line 290. "intermediate carbon fertilisation ..." Not phrased felicitously, since it implies that C-fertilisation itself has multiple levels of working (off, half-on, on). Please rephrase.

Line 299. Sorry, I might have missed this – what do you do about the conversion of secondary vegetation to pasture in terms of carbon. Are we looking at gross or net here? Line 302-303. I am not sure why you are picking out one model from this study. I think you should give the range of estimates. I don't know whether the quoted value for IPSL-CM5A-LR is low-end or high-end. Figure 3. Please don't abbreviate emission on this Figure. Line 305. "probably"? You could establish this by looking at what the difference in loss of non-agricultural land between these simulations and yours is if you take out the pasture expansion and the. expansion of cropland in Alaska. Line 308 et seq. So the difference is caused by the differences in the scenarios, right? But later on you imply its because the models don't include nitrogen-limitation. I think you need to make it clear what you think is giving rise to these differences, scenarios or model set-up. It would, of course, make it interesting to run your experiments with the older scenarios - and this would be helpful in terms of uncertainty analysis. Line 309-310. Your comparisons with other simulations are unbalanced - having described the results from the Brovkin et al study in some detail you here say that the results are low compared to the Nishina et al. (2015) study even when comparing just to the models in that study with nitrogen limitation. But no details. How low? what was the range simulated by the N-enabled models in Nishina et al.? Do you have any idea why you

get a different result? Lines 311-318. So, different models produce estimates less than LandSyMM as well as above. What do we learn from this? You imply this is because of differences in scenarios (while hedging your bets in terms of climate forcing), but what is needed here is a back-of-the-envelope calculation of whether the differences in cropland and pasture area would produce a comparable estimate in LandSyMM. If you wanted to be ultra-realistic, you could use the areas where they show biggest changes in area. Line 319. Photosynthesis scaling parameters ...... what scaling parameters? Line 338. Why did you not run it in coupled mode then? Line 341. Please can you explain what was done in the Asadieh and Krakauer (2017) analyses. Were these full hydrological models? Since this was an ensemble, presumably there is a range of estimates for at-risk of flood and at-risk of drought? Please give these ranges in the text. How much of a difference does using monthly versus daily values make to the estimates of area affected? Line 349. How many classes are there? You need to spell out that you are talking about increases/decreases in flood risk and drought risk areas). Line 379. Please explain why high CO2 suppresses BVOC formation and provide references here. Line 382-383. This sentence is confusing because it seems to say that boreal forests are causing declining monoterpene emissions – whereas I think the idea is that decreases in boreal forest area coupled with less effective BVOC production in the surviving boreal forest area are responsible. Please rewrite. Line 393. Please italicise Miscanthus Line 396 et seq. This paragraph states that predicting the effects of changing BVOCs is difficult because the model framework doesn't include atmospheric chemistry. I am not sure what "a surface-level discussion of possible effects" means here. You can predict potential changes in BVOC emissions, and so perhaps this is the point to stop at. There is no need to go further and speculate about what the impact of these changes might be on atmospheric chemistry, climate and/or health. Line 403. The fact that this one region is not defined as a hotspot, and that it has a big impact on the results, makes a good case for extending this analysis to consider changes in biodiversity everywhere. Line 410-415. Please rewrite this section to first make the comparison and then explain possible sources of differences. Line

415. Are you saying that species-area curves are an inappropriate tool for estimating extinction rates rather than species numbers? Lines 419-420. I hadn't realised that climate changes and CO2 could have an effect on models too! Please rephrase this. Line 423. "We may see a similar effect ...." Please clarify: do you or don't you? Line 429-430. Considering that the results from LandSyMM have been compared to a range of other model simulations in this paper, the first sentence really doesn't make sense. Maybe this is more comprehensive in terms of the range of scenarios and the range of outputs, but what else is different here?

---

## Author Comment (AC1) · 10 Feb 2020

*This article presents an ambitious modelling exercise that combines the LPJ-GUESS dynamic global vegetation model with the PLUM land-use model (now under the new name LandSyMM). While the model has been presented before in Alexander et al 2018, the coupled models have not been presented before from the LPJ-GUESS perspective which makes this an interesting article. Additionally, the large number of ecosystem service indicators make the presented analysis interesting on its own.*

*Before the article can be published in ESD however the structure of the article needs substantial improvement. While I acknowledge the challenge of describing such a complex model in a comprehensive as well as concise manner, I do think improvements can be made. I have three general points and a number of more detailed points that need to be addressed.*

We thank the referee for their helpful comments.
* * *
*The structure of the methods section is confusing. I would recommend to start with an overview of the models used and how they interact (a diagram might be helpful), then a detailed description of the different ecosystem service indicators analysed.*
*Also, I would recommend to reduce the size of the methods section by moving some of the information on detailed input for LPJ-GUESS and PLUM to the SI. On the other hand, much more information is required on the scenario setup. Details about the main assumptions should be presented. A table with an overview would be helpful. It is important that the reader does not depend on a different article to understand how the presented scenarios are defined.*
*The extra experimental scenarios that have been performed to improve understanding of the results are confusing. Maybe these can be added to a descriptive scenario table, either in the main text or the SI, and they could be given explicit names to make it easier to refer to them in the results section, e.g. SSP4-2.6-noCO2/noLUC. I think this is better than for example the long text 'in the constant climate+CO2 experiment' (line 295-296).*

This section has been significantly reworked:
- Sect. 2.1 now focuses on LPJ-GUESS, with the text on ecosystem services having been moved to the Introduction and the new Sect. 2.5.
- Sect. 2.2 focuses (as before) on PLUM, now including some text about where it uses data from and gives data to LPJ-GUESS.
- Sect. 2.3 describes how the coupling works. This is necessarily very technical, but a flowchart figure is now provided for clarification.
- Sect 2.4, describing input data and scenarios, has been compressed significantly relative to the old Sects. 2.3.1–2. Technical information regarding data sources is now less prominent, with about half of the section serving instead to provide context about the SSPs and RCPs. For interested readers, the Supplementary Methods provide more technical detail.
- Sect. 2.5 focuses now solely on the ecosystem service indicators used in the study. Background information on the ecosystem services in question has been moved to the Introduction.

A LandSyMM overview diagram (Figure 1) has been added to Sect. 2.3. Experiment nomenclature has been standardized and is explained in the new Table 1 (Sect. 2.3).

*The results and discussion section is rather lengthy and is (as the names correctly states) a mix between results and discussion. I think this reduces the clarity of your story and is not in line with the standard outline of scientific articles. I think the article would greatly benefit from splitting up this section into a clear description of the results (topic by topic) and subsequently a discussion of the results in the context of the literature (again, topic by topic).*

We agree that it would be more in line with common practice to move such comparisons to a separate Discussion section, but while this works well for most papers, we believe it would not in our case. First, some of our results with regard to land use area are so striking (and different from results from similar work) that it makes sense to address them immediately. This also provides the reader with valuable context for interpreting the rest of our results. Finally, because this paper touches on so many different ecosystem services, it is necessarily rather long. Postponing the literature comparison to a separate Discussion section would necessitate spending space there reminding the reader of our own results; in the interest of keeping this paper as concise as possible, we have chosen to avoid that.

***Detailed comments:***

*Line 9-10: this is the first time biodiversity is mentioned in the abstract, while it is presented as one of the major outcomes.*

We have added "biodiversity" to the list at Line 5.

*Line 15: please rephrase 'larger than today's by anywhere from' to something more concise and more academic, e.g. 'an increase ranging from 1.5 billion to 6 billion'*

This clause now reads "with a population increase by 2100 ranging from 1.5 billion to nearly 6 billion people".

*Line 57-102: this section describes the ecosystem service indicators presented in the article. This should be a separate section. In addition, a lot of the text is introduction to why these indicators are important. I think this reduces the clarity of the methods section which should be a more technical description of the indicators presented in the article. Maybe you can move part of the text (in a more concise way) to the introduction.*
*Line 93-102: I assume the biodiversity indicator is not a standard output of LPJ-GUESS, correct? It is now part of the LPJ-GUESS section which is misleading. It should be more clear that this is calculated based on the downscaled PLUM results.*

Description of the analyzed ecosystem service indicators has been moved to the new Section 2.5. More introductory or background text about the ecosystem services has been moved to the Introduction.

*Line 109-110: The Popp et al 2017 article describes a large number of SSP scenarios from 5 IAM models. Please specify which scenario from which model has been used and preferably refer to a paper that presents the results specifically for this model.*

We now specify that the bioenergy demand comes from the MESSAGE-GLOBIOM model.

*Line 115: this is a very detailed start of this section. Consider restructuring the sentence. Maybe move this entire section to the SI as very detailed information for main text.*
*Line 137-138: a summary of the climate and land use data used should be given in the text, not in the SI. Consider moving this text to the SI and summarizing input data in the main text.*

The relevant text (now Sect. 2.3) has been overhauled, and an overview figure (now Fig. 1) has been added.

*Line 161: make sure your language is more academic. Don't use terms like 'briefly' or 'attempts'.*

In this matter of style, we disagree with the reviewer. We have left in "attempts" because it is an accurate description of what the algorithm does. "Briefly" primes the reader to expect a non-comprehensive explanation of the algorithm, improving reading flow. That said, most of this text has been expanded and moved to the new Supplementary Methods Sect. SM3, and "briefly" has been removed.

*Line 178: what is minimum 'non-agricultural area'. This sounds like a PLUM-specific technical term, please rephrase.*

This has been clarified: "These included input and transport costs, tariffs, and minimum non-agricultural area (which places an upper limit on the total fraction of a gridcell that PLUM can allocate to cropland and pasture)."

*Line 203: 'cropland area expands about 10% between 2050,. . .'. Between 2050 and what?*

This has been corrected to read "between 2050 and 2100".

*Line 214: 'what PLUM calls "other management"'. Please use more academic language and avoid usage of model-specific technical terms.*

We have changed "what PLUM calls" to "PLUM's". However, we have left the reference to "'other management' intensity," since avoiding model-specific terms harms reproducibility. For readers not already familiar with the term, our explanatory parenthetical text—"(representing, e.g., pesticide application)"—should provide sufficient clarification.

*Figure 1: why do you only show ruminants? Non-ruminants also have a very strong effect on the agricultural system due to high feed requirements.*

We originally excluded monogastrics from this figure in the interest of limiting its size, since the role of monogastrics demand is not discussed in the text, and interested readers could find its trajectory for each SSP in what is now Fig. SR2 (global demand trajectories for each commodity; formerly Fig. SR1). However, we have now added monogastrics demand to this bar graph (now Fig. 2).

*Also, I am surprised about the very strong increase in ruminant demands. FAOSTAT actually shows that in recent decades demand for ruminants-based products has increased relatively little while monogastric-based products have increased much stronger.*

Monogastrics do make up a majority of meat supply by raw tonnage:

[Figure]

However, they make up a minority of meat supply once converted to the units of PLUM demand, which are tons feed equivalent: Ruminants require much more feed to produce a given weight of meat than do monogastrics. To convert, we multiply poultry meat, pig meat, mutton/goat meat, and beef respectively by 3.3, 6.4, 15.0, and 25.0 tons feed per ton product:

[Figure]

Additionally, ruminant feed requirements in PLUM include the production of milk, the consumption of which has been increasing more rapidly than ruminant meat. This further lowers the fraction of monogastric products in the feed-equivalent figures (0.7 tons feed per ton milk):

[Figure]

Furthermore, shifts in where demand is growing will change the makeup of the global livestock product landscape. The above figures show that recent trends in livestock meat demand have been driven by increasing consumption in Asia, where meat demand is mostly for monogastrics. In contrast, PLUM projects strong increases in livestock demand in sub-Saharan Africa as population and wealth increase there; that region has historically had a much lower fraction of livestock demand comprised of monogastrics. This is shown in the figures below, which present PLUM-projected demand for Sub-Saharan Africa as compared

with combined China, South Asia, and Indonesia. (Region groupings are different between these plots and previous. Also note that colors here stand for scenarios, whereas colors previously stood for regions.)

[Figure]

PLUM also projects that the fraction of livestock products (meat and milk) provided by monogastrics will decrease as these regions get wealthier:

[Figure]

*Line 243: I was taught to avoid the term 'forecast' (sounds like something a fortune teller would say) but rather use the term 'project', but maybe this is a personal preference.*

This has been changed to "projects".

*Line 251-252: this sentence is extremely vague and unhelpful. Please make more explicit.*

The last two sentences of this paragraph have been changed to the following: "While we do not expect LandSyMM's results to necessarily match those of other models, such a large, qualitative difference requires explanation. Several factors related to experimental setup and overall model structure likely contribute."

*Line 276: it is impossible to have 500% or 700% higher land use, using current agriculture ($\sim$50 Mkm2) this would mean 250 or 350 Mkm2 which is more than the terrestrial area of the world.*

These values refer to *relative* difference from the Alexander et al. (2018) differences. That is, the range of cropland area among scenarios in the present work is six times what the range was in Alexander et al. (2018). The text has been changed to read (new text in **bold**): "The **spread** in land-use area **projections** between the most extreme scenarios is much higher in this work than…"

*Line 282: what do you mean with infrastructure efficiency? Also, if I understand correctly you make similar SSP-specific assumptions in PLUM (SM6). These should also lead to a higher spread in land-use projections, right?*

PLUM does make some SSP-specific assumptions, but not the ones described in this sentence. The sentence has been changed to read (new/amended text in **bold**):

> As described above, IMAGE makes a number of assumptions (based on the SSP storylines) **that PLUM does not** regarding future deviations from historical 'business-as-usual' trends and relationships, including dietary shifts, **reductions in food losses during transport**, and forest conservation.

*Line 276-278: I don't see why the RCPs lead to a much larger spread in scenario results?*

After the cited sentence we have added the following: "The wide variation among the SSPs in population and economic growth trajectories, along with SSP-specific PLUM parameters (Sect. 2.3.2), contribute to this increased spread."

*Also, is PLUM informed about the yield effects of climate change as these would impact for example trade and food security?*

Informing PLUM about the yield effects of climate change (and changing $CO_2$ concentration) is indeed the reason we feed it with LPJ-GUESS-simulated potential yields. Text has been added throughout the manuscript to emphasize this point.

*Line 325: is this realistic? Could not reduced feedback effects from lower evapotranspiration from forests in fact reduce runoff?*
*Line 336-337: how can increased agriculture reduce the risks on droughts? Please explain.*

The end of the paragraph originally ending with Line 326 has been changed to read (new text in **bold**):

> Deforestation in central Africa, for example, is the primary driver of increasing mean annual runoff there **because of reduced evapotranspiration relative to existing vegetation. Note, however, that LandSyMM can only represent the effect of land cover change on evapotranspiration and runoff directly—to include the impact of these flux differences on rainfall would require a coupling with a climate model.**

We have also added the following text in the Methods: "The CMIP5 runs did include land-use change, but not the trajectories output by PLUM. As such, and as with all models that are not climate-coupled but rather use offline forcings, we do not consider the effects of our simulated land-use change on climate."

*Line 348-349: I do not understand what the 'fraction of included land area' means and why it shows that not including routing is not a big issue (it sounds disconcerting to me). Please explain better.*

In that sentence, "the fraction of included land area in any class" has been changed to "the results for any class".

*Line 359-360: why is the estimate more consistent if there is less over application of N? In reality many countries exceed the N uptake rate of plants (most notably China). It sounds less realistic to me that N application only increases by 2%. Does this not imply a major break with historical trends?*

The integrated assessment models whose output was used in Krause et al. (2017) actually simulated these very high levels of nitrogen input as resulting in very high yields—i.e., they did not simulate the real-life phenomenon of overapplication. We do not attempt to account for overapplication, either, but in Alexander et al. (2018) we showed that LandSyMM nevertheless does reproduce historical N application levels globally.

To clarify our point here, we have changed the text to read (edited text in **bold**):

> … used fertilizer information from IMAGE and MAgPIE**. Strong increases in fertilizer in those models resulted in strongly increased yields, but nitrogen limitation is alleviated at much lower levels in LPJ-GUESS. IMAGE and MAgPIE fertilization rates thus** often exceeded what plants in LPJ-GUESS could actually take up, resulting in high amounts of N loss. Coupling LPJ-GUESS with PLUM provides for a more internally consistent estimate of future N losses**, while still reproducing historical fertilizer application well (Alexander et al., 2018)**.

*Line 369-370: you cannot state 'and other models' here if you refer to three articles that are if I am not mistaken all based on LPJ-GUESS. It is quite logical then that the estimates are similar. Please add an independent reference.*

We have replaced this sentence with the following (edited/new text in **bold**):

> Global combined BVOC emissions over 2001–2010 totaled ~546 TgC yr$^{-1}$ (~503 and ~43 TgC yr$^{-1}$ for isoprene and monoterpenes, respectively), which compares well with estimates from LPJ-GUESS **using different land use scenarios (Arneth et al., 2008; Hantson et al., 2017; Szogs et al., 2017) and the MEGAN model (Sindelarova et al., 2014).**

*Line 430: I don't think 'storylines' is the right word here as you don't calculate storylines but scenarios that are described by a certain storyline.*

We have replaced "storylines" with "scenarios" there. To avoid repetition, in the previous sentence we replaced "scenarios" with "possible futures".

*Line 435-439: would it be possible to draw stronger conclusions based on the scenario assumptions on how certain future developments should rather be avoided etc?*

To maintain objectivity, we have decided to avoid such prescriptivist language.

**SI:**

*Figure SR2: why are the starting points in 2010 so different for irrigation and fertilizer? This should be historic data I assume? Also, there is hardly any trends in the irrigation results, why is this the case?*

This figure (now Fig. SR3) presents PLUM outputs pre-harmonization; we have added text to this effect in the caption. These raw PLUM outputs are not necessarily expected to align exactly with historical data, as evidenced by the need for the harmonization routine. Nor are they necessarily expected to align with each other at the beginning of the period, because of scenario-specific parameters in PLUM derived from the SSPs—particularly regarding the cost of irrigation and fertilizer. Text to this effect has been added to Sect. 2.2 and the caption of what is now Fig. SR3.

If PLUM were to begin with some "historical" parameter values and gradually phase in scenario-specific values, this would improve agreement among scenarios at the beginning of the future run. However, as there is no obviously "correct" way to design this phase-in, we have made the parsimonious decision to apply all scenario-specific parameters at the beginning.

Regarding irrigation, the following text has been added to the end of the second paragraph of Sect. 3.1:

> PLUM prescribes lower irrigation rates by the end of the century for most scenarios (Figs. 2, SR2). This is enabled by higher global mean rainfall in all RCP scenarios, as evidenced by the bars for runoff in Figure 4, as well as by improved water-use efficiency for crops other than $C_4$ cereals due to increased $CO_2$ concentrations. Crop demand increase in SSP3-60 outweighs these effects, however, resulting in higher irrigation in that scenario.

*Figure SR3: I don't understand the top figure on livestock demand. The order of magnitude makes it likely the results are on crop production, but the title suggests this is total production (?) of livestock products. But the bottom figure shows feed for livestock so if the top figure is also about feed demand for livestock it would not be useful. Please explain and improve.*

The former Fig. SR3 (now Fig. SR4) has been simplified to show only the fraction of ruminant food that is provided by feed crops, which is the only information required to support the corresponding assertion in the main text (i.e., that feed becomes much more important in raising ruminant livestock beginning around 2090).

*Figure SR8: similar figures are shown for south asia and sub-saharan Africa. Why not consistently show results for all regions? Maybe in slight smaller panels and without the figures with maps in between? Also, why is this figure shown as a delta instead of absolute amounts and how can it be that the demand is so extremely jumpy for oil crops? This seems very unrealistic.*

There are an enormous number of possible regions and region groupings that could have such figures made; we present only what is necessary to support specific assertions made in the main text. Similarly, we have presented percentage change rather than absolute amount because the former is more directly explanatory of assertions in the main text. Showing absolute amount would, in some cases, make relevant trends difficult to discern.

The "jumpiness" of individual crops is due to shifts in which crops are used as animal feed. These shifts are due primarily to changes in relative prices of the different crop commodities. Note that the dotted lines, which exclude demand for animal feed, are much more stable. It is indeed unrealistic to expect, e.g., oilcrop production to triple from one year to the next, as would be required to satisfy the demand increase seen in the US and Canada in the early 2040s (Fig. SR9, formerly Fig. SR8). For the purposes of our ecosystem services analysis, however, gross decadal trends in total agricultural area and management inputs are much more important than exactly what is being grown on cropland, and those gross trends are much smoother. Text explaining this has been added to Sect. 2.2.

*Figure SR10: please write complete description instead of referring to another figure.*
Complete descriptions have been added to the former Figs. SR 5–7, 10, and 13 (now SR 6–8, 11, and 14).

---

## Author Comment (AC2) · 10 Feb 2020

*The main contribution of this paper is in coupling the PLUM and LPJ-GUESS models to project land-use change impacts in future scenarios, in terms of biodiversity impacts or greenhouse gas emissions.*

*As such, my main criticism of the paper is that the method section is not very detailed about (1) the assumptions of the two models, (2) the working of the two models, and most crucially (3) how these were combined. While I appreciate the difficulty of communicating complex models in a brief section, seeing that this is the central contribution of the paper, the reader should not be forced to go through the supplemental materials (which is also very densely presented) to understand the models and their interplay. This could potentially be presented as multiple tables and a joint figure exploring the interactions and basic properties of the models.*

Section 2 has been extensively reworked:
- Sect. 2.1 now focuses on LPJ-GUESS, with the text on ecosystem services having been moved to the Introduction and the new Sect. 2.5.
- Sect. 2.2 focuses (as before) on PLUM, now including some text about where it uses data from and gives data to LPJ-GUESS.
- Sect. 2.3 describes how the coupling works. This is necessarily very technical, but a flowchart figure is now provided for clarification.
- Sect 2.4, describing input data and scenarios, has been compressed significantly relative to the old Sects. 2.3.1–2. Technical information regarding data sources is now less prominent, with about half of the section serving instead to provide context about the SSPs and RCPs. For interested readers, the Supplementary Methods provide more technical detail.
- Sect. 2.5 focuses now solely on the ecosystem service indicators used in the study. Background information on the ecosystem services in question has been moved to the Introduction.

*Conversely, I would suggest to shift large parts of the input data sections to the SM, as (especially in the case of PLUM), these are largely technical details on the modeling side. Instead, the manuscript should spend more time in detailing the scenario setup, as well as how the "holding constant of certain variables" for the purpose of robustness checking was implemented, as based on the abstract and introduction this is a central part of the paper.*

Technical details are significantly less prominent in the new Sect. 2.3. The new Table 1, which describes the experimental runs, should clarify what it means for certain variables to be "held constant."

**Minor comments**

*The error bars in Fig. 1 and Fig. 3 are largely cosmetic, as the processes depicted here are highly persistent (e.g. population, cropland), and the error bars merely measure the standard deviations within a decade. The authors themselves do not interpret them within the text, so for the clarity of information they could be also left off. Indeed, if the authors would like to highlight the temporal dynamics, a representation of the whole time-series would be better suited.*

The error bars in these figures (now Figs. 2 and 4) have been removed.

*In Fig. 2 it would be good to either have a different color scheme for the two columns or the same scale.*

This figure (now Fig. 3) has been updated. Among other visual changes, the two columns are now on the same scale.

*In the SM, figures for commodities and exports are presented (SR8, 10,11, 13). Here the trade patterns exhibit highly cyclical behavior, which right not be fully realistic. This should be contrasted with past export dynamics in the same crops and regions.*

The "jumpiness" of individual crops is due to shifts in which crops are used as animal feed. These shifts are due primarily to changes in relative prices of the different crop commodities. Note that the dotted lines, which exclude demand for animal feed, are much more stable. It is indeed unrealistic to expect, e.g., oilcrop production to triple from one year to the next, as would be required to satisfy the demand increase seen in the US and Canada in the early 2040s (former Fig. SR8, now SR9). For the purposes of our ecosystem services analysis, however, gross decadal trends in total agricultural area and management inputs are much more important than exactly what is being grown on cropland, and those gross trends are much smoother. Text explaining this has been added to Sect. 2.2 (PLUM).

---

## Author Comment (AC3) · 10 Feb 2020

*Excellent attempt to evaluate scenarios for implications on ecosystem services including trade-off and co-benefits. LandSyMM represents state-of-the-art in modelling of the land-use & vegetation. This type of work is extremely relevant for ecosystem service provision in the context of the Paris Agreement (food-energy-water nexus and beyond). Follows on from earlier studies, e.g. Krause et al., 2017.*

*Overall I enjoyed reading the manuscript and warrants publication. I found it very informative and represents a substantial amount of work and scientific progress. A minor edit is warranted in the methods section which I found confusing in places. It could be improved as it was difficult to work out exactly how the models link together and finally the number of runs made. The authors also need to clearly distinguish general text explaining processes from what's actually included in the models. I found the results/discussion section comprehensive and complemented with extremely informative figures. Interesting new advances on attempting to relate flood/drought (see specific comment below), and the link to land loss in biodiversity hotspots.*

We thank Prof. Sitch for his kind comments and helpful suggestions. Section 2 has been extensively reworked:

- Sect. 2.1 now focuses on LPJ-GUESS, with the text on ecosystem services having been moved to the Introduction and the new Sect. 2.5.
- Sect. 2.2 focuses (as before) on PLUM, now including some text about where it uses data from and gives data to LPJ-GUESS.
- Sect. 2.3 describes how the coupling works. This is necessarily very technical, but a flowchart figure is now provided for clarification.
- Sect 2.4, describing input data and scenarios, has been compressed significantly relative to the old Sects. 2.3.1–2. Technical information regarding data sources is now less prominent, with about half of the section serving instead to provide context about the SSPs and RCPs. For interested readers, the Supplementary Methods provide more technical detail.
- Sect. 2.5 focuses now solely on the ecosystem service indicators used in the study. Background information on the ecosystem services in question has been moved to the Introduction.

**Minor comments:**

*28% nitrogen pollution -> aquatic systems, air pollution? Unclear from line 75-90 how nitrogen pollution is quantified/considered in the LPJGUESS (this section is more an introduction to N pollution)*

The following sentence has been added to Sect. 2.5: "This is the combined rate of loss from dissolved N leaching (a function of percolation rate and soil sand fraction), denitrification (1% of the soil mineral nitrogen pool per day), and fire."

*Line 24 "As global environmental and societal changes continue to accelerate over the coming decades,". Somewhat vague. Are we sure changes are and will accelerate.*

We have deleted "to accelerate" from the sentence in question.

*LandSyMM represents state-of-the-art in modelling of the land-use & vegetation. However this advance is not entirely clear. Lines 35-40 could be improved to demonstrate the advance beyond existing IAMs. e.g. "is unique among global land-use change models in the high level of spatial detail that it considers in the response of agricultural yields to management inputs, as well as in allowing short-term over and under-supply of commodities relative to demand (rather than assuming market equilibrium in every year)". Vague – what does high-level of spatial detail mean? I wonder if a table contrasting CMIP6 IAMs with LandSyMM would be useful.*

The text of the last paragraph in the Introduction has been modified to read (new/edited text in **bold**):

> … This coupled model system—the Land System Modular Model, or LandSyMM—is **among the state of the art in global land-use change models due to the high level of detail that it considers in the response of agricultural yields to management inputs. Whereas most integrated assessment models rely on generic responses of yield to changing climate, atmospheric carbon dioxide, and fertilizer, LPJ-GUESS simulates these processes mechanistically. Land use optimization also happens at a finer grain in LandSyMM (about 3400 gridcell clusters) than in other similar model systems (tens to hundreds of clusters). Finally, LandSyMM is unique in that PLUM allows** short-term over- and under-supply of commodities…

*Also perhaps a real world example can be given to demonstrate the importance of the non-equilibrium assumption (i.e. is this a detail or fundamental). Perhaps the authors can elaborate more on the differences between PLUM and other LU model approaches in section 2.2.*

A new paragraph has been added to Sect. 2.2 (PLUM) explaining the significance and highlighting the novelty of the non-equilibrium assumption in PLUM (new text in **bold**):

> To solve for land use areas and inputs that satisfy demand, PLUM uses least-cost optimization, which allows for short-term resource surpluses and deficits. **Such imbalances can be significant in the real world: Global supply of major cereal crops frequently swings 5 to 10% out of equilibrium on an annual aggregate basis, and more extreme imbalances can be seen at the scale of individual countries (FAOSTAT, 2018a). These dynamics are not captured by equilibrium models, such as those used in other land use and integrated assessment models, which represent for each year the stable state that the economic system would move to eventually if the environment did not change. Because global agricultural markets are not in equilibrium, disequilibrium models are needed to capture the real-world process of moving towards---but not reaching---equilibrium in a constantly-changing economic and physical environment. Disequilibrium models have received varying amounts of attention in the literature over time (e.g., Kaldor, 1972; Mitra-Kahn, 2008; Arthur, 2010), and to our knowledge PLUM is the first land use model to incorporate one.**

*Very good coverage of LPJGUESS, however somewhat generic in places. The authors should explicitly state which metrics are calculated for ecosystem services, e.g. how is nitrogen pollution calculated? (at first it gives the impression of some combination of health, water quality, air pollution impacts on vegetation indices; are these processes modelled in LPJGUESS?), i.e. this text gives a more general description of nitrogen pollution, rather than what is actually modelled in LPJGUESS.*

The following sentence has been added to Sect. 2.5: "This is the combined rate of loss from dissolved N leaching (a function of percolation rate and soil sand fraction), denitrification (1% of the soil mineral nitrogen pool per day), and fire."

*Good attempt, but I'm not entirely convinced of the approach for changing flood risk based on monthly runoff – I think this should be raised again in the discussion as a potential limitation/uncertainty.*

    The following text has been added to Sect. 2.5:

As Asadieh and Krakauer (2017) note, these metrics do not translate directly into impacts due to the mitigation capacity and nonlinear effectiveness of reservoirs, flood control mechanisms, and other infrastructure, as well as changes in demand and mean climate. However, changes in streamflow extremes have served as rough indicators in a number of previous global-scale studies (e.g., Tang and Lettenmaier, 2012; Hirabayashi et al., 2013; Dankers et al., 2014; Koirala et al., 2014).

The following text has been added to Sect. 3.2.2:

As discussed in Sect. 2.5, these values are not direct measurements of flooding or drought impacts, but they do serve as useful indicators.

*As noted by the authors they apply the Asadieh and Krakauer 2017 approach but use monthly surface runoff data. The authors should state the temporal resolution the model is applied (I assume monthly?). As I understand LPJGUESS outputs per gridcell 1) monthly total N Loss 2) monthly runoff 3) annual land conversion in hotspots, and then interpreted in terms of nitrogen pollution (but no additional metrics used), flood/drought risk (using AK 2017 metric), and loss of land in biodiversity hotspots.*

    We have added the following to Sect. 2.1 (LPJ-GUESS): "Hydrological and most physiological processes are modeled at daily temporal resolution; vegetation growth, establishment, disturbance (including land-use change), and mortality happen annually."

*Somewhat confusing section 2.3 on simulation details. For example, Line 115-120 "In each scenario, every grid cell is planted with each crop type, each of which is given six different management treatments in a factorial setup: fertilization of 0, 200, and 1000 kgN ha–1 and either no irrigation or maximum irrigation." Why? Somewhat comes out of the blue.*

    Text has been added ("under a range of irrigation-fertilization treatments") to Sect. 2.2 to introduce the idea of this setup.

*Then in 2.3.3 experimental setups Lines 180-190 "In addition to the LPJGUESS runs forced with PLUM-output land use and management trajectories (harmonized as described in Sect. 2.3.1), six experimental runs were performed for each scenario, to disentangle the direct effects of climate change (including CO2 concentration increases) from those of land use and management change." This is another 6 simulations?*

    Yes, those 6 are different runs (as opposed to treatments within one run). This is hopefully clarified by the overhaul of the Methods section.

*Then there's PLUM-forced runs. So I assume this is not the standard approach to running LandSyMM (i.e. PLUM coupled to LPJGUESS), i.e. how frequent is information from LPJGUESS and PLUM exchanged (annually, e.g. potential yields line 167?) Perhaps consider a table with a full list of the simulations would help the reader.*

The new Sect. 2.3 is now clearer in describing that the PLUM-forced runs are part of the way that LandSyMM is designed to be run. The fact that the LPJ-GUESS to PLUM coupling happens at five-year timesteps is now mentioned (first sentence, third paragraph of new Sect. 2.3). A table of simulations has been added (Table 1), and Table SM1 provides additional clarification.

*Line 140 "The calibration run was forced with climate data from CRU-NCEP version 7 (Le Quere et al., 2016), but with CRU TS3.24 precipitation (Harris et al., 2014) due to problems discovered in the CRU-NCEP precipitation data" What were the problems? (others are using CRUNCEP7 so would gain from this information)*

This is now explained in a footnote in Supplementary Methods Section SM1: "The CRU-NCEP algorithm was designed to match CRU TS3.24 monthly precipitation totals, but it produced unrealistically high numbers of wet days—days with precipitation of at least 0.1 mm—in the tropics and boreal regions in the early part of the 20th century."

*Line 143 IPSL-CM5A-MR – what's the climate sensitivity of the IPSL model? Why was this selected? (what are the characteristic features of this GCM future prediction), for example which areas are projected to have higher / lower precipitation, as this will govern the simulated flood/drought risk (and affect the other ecosystem services studied?). For example, around line 327 perhaps add some text / speculate on impact of using one climate model for regional runoff. E.g. Figure 5, perhaps it would be useful to include a map of change in precipitation (and temperature) to give the reader a feeling for the importance of choice of GCM.*

Ahlström et al. (2012) looked at 18 CMIP5 climate model outputs and used them to force LPJ-GUESS. We used that analysis as a guide to selecting which model's forcings we would use. Initially, we wanted to use MPI-ESM-LR, as it represented a middle-of-the-road in terms of both mean land temperature rise and net ecosystem exchange response in LPJ-GUESS (Ahlström et al. 2012, Fig. S3). However, the MPI-ESM-LR outputs were not available for RCP6.0, which we needed for two of our RCPs. We instead chose IPSL-CM5A-MR, which had all RCPs available and is on the low side of the high end in terms of mean global temperature change. However, IPSL-CM5A-MR does simulate a large increase in precipitation around the Equator (ibid., Fig. S2).

Bars for global precipitation change have been added to what is now Fig. 2. This is now referenced at the beginning of Sect. 3.2.2. Additionally, a figure showing maps of mean change in temperature and precipitation for each RCP in this study has been added to the Supplementary Results. A reference to this figure has been added to the Methods (Sect. 2.4).

The following text has been added to the end of Sect. 3.1.1:

> Krause et al. (2017) used climate forcings from the IPSL-CM5A-LR model, which differs from what we used (IPSL-CM5A-MR) only in that the former was run at a lower resolution. Both have similar mean global land temperature changes: for RCP8.5, on the low side of the high end of 18 CMIP5 models examined in Ahlström et al. (2012). This temperature change is strongly correlated with net ecosystem carbon exchange (land-to-atmosphere C flux, excluding fire emissions), so a different choice of climate forcings could have resulted in a stronger C sink or even a C source (Ahlström et al. 2012, Fig. S3).

The following text has been edited in Sect. 3.2.2 (changed text in **bold**):

> Some differences between our results and those of Asadieh & Krakauer (2017) might be expected because we used monthly instead of daily values for P95. **Also, whereas that study used five climate models, we used only one---specifically, one that simulates a much larger precipitation increase around the Equator (in RCP8.5) than 18 other models examined in (Ahlström et al. 2012, Fig. S2). Finally,** LPJ-GUESS is not a full hydrological model: e.g., it does not include river routing.

*Line 146 onwards. "Time-evolving historical land use fractions—i.e., the fractions of land in each gridcell that are natural vegetation, cropland, pasture, or barren—were taken from the Land Use Harmonization v2 dataset (LUH2; Hurtt et al., in prep.). The MIRCA2000 dataset (Portmann et al., 2010) provided crop type distributions for the year 2000, which were used for all historical years." I'm a bit confused what is used here. So LUH2 give the cropland coverage, but not which crops, that's given from MIRCA2000 and relative proportion of individual crop types stay fixed through time over the historical period?*

Yes, that's the correct interpretation. This should now be clarified in text at the end of the new Sect. 2.4.

*Lines 168 onwards describe the SSPs. I think this generic text would work better if it came earlier (SSPs have already been mentioned in several places already).*

As part of the Methods section overhaul, the SSPs are now introduced at a more appropriate point.

---

## Author Comment (AC4) · 10 Feb 2020

*The paper presents and explores simulations with the coupled land use and vegetation model LandSyMM to quantify future land use change and resulting impacts on ecosystem service indicators. There is a lot of interesting and thought-provoking material here, and I am sure this paper will create a lot of interest. However, like many "future scenario" papers, there is a lack of consideration of plausibility or uncertainty. The authors do not help the reader to understand why these projections are better or more reliable than other estimates. The section on runoff and flood risk is not convincing, in part because the separation of responses takes no account of what is already known about impacts of CO2 changes on runoff and in part because no case is made for using mean P95month as a measure of flood risk in a model with no water redistribution instead of relying on a projections using explicit hydrological modelling. The writing style is overblown (particularly in the Introduction) and often obscure (for example in the methods section and in the results section). The messages here could be conveyed in a clearer fashion with some pruning and rewriting, and this would considerably improve the readability of this paper. Shortening the existing text would leave room for a proper discussion section that would allow key issues to be explored. I hope the specific suggestions below can help the authors improve this paper and clarify their arguments, because the reliable estimation of future changes in ecosystem services is important for many purposes and people.*

We thank Prof. Harrison for her detailed and helpful comments.

**Specific Comments**

*The Methods section is long, difficult to follow and at the same time does not give sufficient information to allow these experiments to be repeated. I think this needs rewriting, focusing on the information that is really needed to understand what is going on.*

> This section has been significantly reworked:
> - Sect. 2.1 now focuses on LPJ-GUESS, with the text on ecosystem services having been moved to the Introduction and the new Sect. 2.5.
> - Sect. 2.2 focuses (as before) on PLUM, now including some text about where it uses data from and gives data to LPJ-GUESS.
> - Sect. 2.3 describes how the coupling works. This is necessarily very technical, but a flowchart figure is now provided for clarification.
> - Sect 2.4, describing input data and scenarios, has been compressed significantly relative to the old Sects. 2.3.1–2. Technical information regarding data sources is now less prominent, with about half of the section serving instead to provide context about the SSPs and RCPs. For interested readers, the Supplementary Methods provide more technical detail.
> - Sect. 2.5 focuses now solely on the ecosystem service indicators used in the study. Background information on the ecosystem services in question has been moved to the Introduction.

*I think it might be helpful to provide a paragraph at the beginning of this section to explain the logic of the order of presentation – I found some information I expected in one section in somewhere else completely, for example.*

Hopefully the reorganized Section 2 will avoid such issues.

*Some of the information presented could be summarised in the form or a table and/or flowchart diagram, and this would certainly be helpful.*

A flowchart is now included, as is a table describing the various experimental runs.

*Section 2.1 LPJ-GUESS description. Given how important these simulations are for downstream results, it would be helpful to give a more detailed description of how the model simulates crops (i.e. what are the differences between the treatments of each crop type), how nitrogen limitation is handled, what information is used to specify nitrogen inputs to cropland etc. The information about how irrigation, water demand, water supply, and plant water stress are simulated may well be described in Alexander e al. (2018) but since these are crucial to the current simulations, the approach should be briefly described here. Even the description of how the model simulates natural vegetation types is given short shrift here, so that the claim that it handles CO2 fertilisation is unsupported.*

In a revised version, we will add information about the performance of LPJ-GUESS relative to other dynamic global vegetation models with regard to primary production, $CO_2$ fertilization, and nutrient limitation. We will also briefly describe how N limitation and irrigation work in LPJ-GUESS. The fertilizer input datasets are described in the revised Sect. 2.4.

*It is also unclear from the present description how some of the service "proxies" are calculated by the model. For example, how does LPJ-GUESS simulate runoff? Please provide a better description of how the model works, so that it is easier to understand its strengths and limitations.*

The revised Sect. 2.5, excerpted here, better describes how LPJ-GUESS simulates some of the ecosystem service indicators we use:

> LPJ-GUESS simulates a number of output variables that here serve as the basis for quantifying ecosystem services. The carbon sequestration performed by terrestrial ecosystems is measured as the simulated change in total carbon stored in the land system, including both vegetation and soil. Ecosystem nitrogen in LPJ-GUESS is lost in liquid form via leaching (a function of percolation rate and soil sand fraction), and in gaseous form through denitrification (1% of the soil mineral nitrogen pool per day) and fire. Here we combine these into a value for total N loss. LPJ-GUESS also simulates the emission of isoprene and monoterpenes—the most prevalent BVOCs in the atmosphere (Kesselmeier and Staudt, 1999)—and accounts for three important factors regulating their emission: temperature, $CO_2$ concentration ($[CO_2]$), and changing distribution of woody plant species due to climate and land use change (Arneth et al., 2007b; Schurgers et al., 2009; Hantson et al., 2017).
>
> LPJ-GUESS simulates basic hydrological processes such as evaporation, transpiration, and runoff. The latter is calculated as the amount of water by which soil is oversaturated after precipitation, leaf interception, plant uptake, and evaporation.

*Section 2.1 Ecosystem services. Most of Section 2.1 is given over to a description of ecosystem services. What I was expecting here was information about what model outputs were used as indicators of specific ecosystem services. However, much of the text describes why a particular service is important – which should have been information provided in the introduction and indeed partly is provided there. The description of the simulated index is brief and uninformative. What I think would be more helpful would be to reshape this in the form of a table, listing the service and the model output (or outputs). This would save some space which could usefully then be used to provide more details in the model description so that it is clear how these outputs are obtained.*

The new Section 2.5 is focused solely on how ecosystem service indicators are calculated. We have not provided a table, but hopefully the information should now be well-organized enough that one is not necessary. Background information on ecosystem services has been moved to the Introduction.

*Section 2.2. Description of PLUM. Although a detailed explanation of the model is given in Alexander et al. (2018), it would be really nice to know a little more about it here. In particular, I am intrigued about the interface between the two models. What is the handshake, for example, between the four crop types in LPJ-GUESS and the seven crop types in PLUM? This is not explained here, nor is it explained in the description of the simulations.*

The flowchart (Fig. 1) now points to this information, which is in the Supplementary Methods. Since this is rather technical model detail that has been covered previously (Alexander et al., 2018), we have decided not to put this information in the main text.

*I do not understand how the crop demand optimisation works, and in particular whether this involves considering surpluses and surplus distribution (which should affect commodity prices) or whether it is assumed that there is always a surplus.*

Text in the Introduction explains that PLUM "allow[s] short-term over- and under-supply of commodities relative to demand (rather than assuming market equilibrium in every year)." Text has been added to Sect. 2.2 saying that PLUM allows for short-term resource surpluses and deficits, and explaining the importance and novelty of this feature.

We consider other information regarding the optimization overly technical for most readers; those interested can find complete descriptions in the previous works cited in Sect. 2.2.

*Section 2.3. Given the complexity of the experimental design, the complicated linking of different models, and the multiple sources of inputs, I think it would be extremely helpful for the reader if you included some kind of flow chart here to guide us through.*

A flowchart is now included.

*Section 2.3.3. The factor separation experiments are not well designed. Recycling 30 years of climate is not equivalent to a constant climate. As the results of the FireMIP experiments*

*show, it is difficult to compare these constant climate experiments with constant other experiments when the constant other is based on a single year.*

Sect. 2.3 now explains, in the text as well as a footnote of the new Table 1, what actually goes in to the "constant-climate" run. New text in **bold**: "By holding either climate, atmospheric $CO_2$, or land use and management constant **(or for climate, looping through 30 years of temperature-detrended historical forcings)** over 2011–2100, …" While we acknowledge that looped climate such as we used can introduce artifacts that would be avoided by a random-sampling approach, we believe that clearly explaining this distinction would require too much space and would be overly technical.

*Furthermore, the value of treating all climate variables as a single input seems a bit odd when thinking about productivity – it would be more interesting to diagnose what aspects of climate are crucial. In any case, a better factor-separation approach is needed. Alternatively, given that. these results are "mostly not presented" (line189) you might leave this out.*

An experimental design to separate the influence of different climate variables would add some rigor, but it would also entail many more model runs, as well as the generation of new climate input datasets for LPJ-GUESS. We thus consider it beyond the scope of the present study.

*Results and Discussion section. There is a lot of detail here, but the selection of things to highlight seems somewhat arbitrary. This is particularly the case in the delineation of geographic areas (what, for example, is meant by South Asia?). I was, for example, somewhat surprised by the lack of commentary on changes in China. Given that these kinds of assessments are of largely political interest, I wonder whether there should be some refocussing here – away from biggest changes to most important regions?*

"South Asia" is now defined. If asked to submit a revised manuscript, we will add a bullet point discussing China, as befits its geopolitical importance.

*Some thought should also be given to tabulating results.*

The values provided next to the bars in the two bar graph figures are intended to serve this function while saving space relative to what would be required for a separate table.

*I would strongly advise separating out the Results from the Discussion, creating a separate section. There are many issues affecting the results presented here, including the impact of methodological uncertainties, that really need to be discussed more fully in this paper. I am not suggesting that these issues invalidate the study, but I think it would be helpful to discuss the sources of uncertainty and I suggest that you add a Discussion section, where you can do this.*

- *How sensitive are the results to specific inputs?*

    We considered a comprehensive evaluation of uncertainty related to climate model choice and PLUM parameter selection to be beyond the scope of this study.

- *what is the impact of mixing static and time-varying inputs?*

We acknowledge that looped climate such as we used for the "constant-climate" experiments can introduce artifacts that would be avoided by a random-sampling approach. However, we believe that exploring the possible impacts of this methodology would take too much space in an already lengthy paper, and in any case could not be properly quantified without additional model runs.

- *given that there are large differences between vegetation models in terms of their predictions, how reliable are the LPJ-GUESS productivity estimates? or perhaps, where are they situated with respect to other models? and how much does this matter to the final assessment? ... A second issue that could usefully be included is "CO2 fertilisation" – given that this still appears to be controversial, that there is confusion about this is photosynthesis or WUE, that different models produce different strengths of fertilisation and so on.*

  In a revised version, we will add information about the performance of LPJ-GUESS relative to other dynamic global vegetation models with regard to primary production, $CO_2$ fertilization, and nutrient limitation.

- *How serious is the mismatch between PLUM outputs and the scenarios? How much of an impact does this have on the projections?*

  In a revised version, we will add a few sentences to the results explaining that the harmonization causes strong changes in the PLUM land-use area maps in only a few regions, and most of those discrepancies are reduced dramatically by the end of the century. We will also add a figure to the Supplementary Results illustrating this.

*One additional issue that could usefully be included in the Discussion, but certainly needs to be treated somewhere, is the assumption that increased fertilisation will always produce an increase in production rather than a saturating relationship, shown by analyses of field data.*

LPJ-GUESS actually does simulate, and PLUM does assume, yield as a saturating function of fertilizer application. This is now mentioned in the first paragraph of Sect. 2.2: "PLUM assumes that irrigation and fertilizer produce diminishing returns, such increasing them increases yield at low intensity levels, but less and less so at higher levels, approaching a yield asymptote."

*In comparing LandSyMM results with other models, it would be useful to include a discussion of the. plausibility (or otherwise) of their/your assumptions. This would also deal with the questions: given that there are other simulation results, what does this paper add? and why should we believe the results are more plausible?*

The text of the last paragraph in the Introduction has been modified to highlight advantages of LandSyMM relative to other model systems. It now reads (new/edited text in **bold**):

"…. This coupled model system—the Land System Modular Model, or LandSyMM—is **among the state of the art in global land-use change models due to the high level of detail that it considers in the response of agricultural yields to management inputs. Whereas most integrated assessment models rely on generic responses of yield to changing climate, atmospheric carbon dioxide, and fertilizer, LPJ-GUESS simulates these processes mechanistically. Land use optimization also**

**happens at a finer grain in LandSyMM (about 3400 gridcell clusters) than in other similar model systems (tens to hundreds of clusters). Finally, LandSyMM is unique in that PLUM allows** short-term over- and under-supply of commodities…"

*I would seriously consider taking out the section 3.2.2, but in any case it needs rewriting. Runoff. The impact of CO2 on runoff is going to be strongly dependent on whether we are talking about semi-arid regions or not, and there is now considerable literature on this (which should be cited). I think a more logical way to organise this section would be around climate regions.*

While it is true that $CO_2$ impacts on runoff are strongly regionally-dependent, we feel that describing its effects in our results for each climate region would require too much space relative to this issue's importance to this study. In a revised version, we will add some brief text and citations acknowledging the regional variation in the $CO_2$-runoff relationship.

*The transition from global runoff increasing to "flood" and "drought" risk is abrupt and it would be helpful to actually explain regional patterns of runoff change first. The fact that LPJ-GUESS is not a proper hydrological model, i.e. it does not transfer water between grid cells, it does include groundwater recharge, it does not include surface storage etc. etc. is mentioned in passing here (line 345). But this is a key issue about what "runoff" means and what "flood risk" means. This has been alluded to earlier on by referring to meteorological flood/drought, but it potentially very mis-leading – not for the immediate readers of the paper but certainly for the "assessments" that will pick these results up and re-use them.*

While LPJ-GUESS is not a full hydrological model, its predecessor model LPJ has been shown to perform comparably to such models at the basin scale, at least at the time of publication of Gerten et al. (2004). Since the simulation of runoff in LPJ-GUESS has not changed significantly since then, we feel confident enough in our results at the basin scale to leave this section in. However, we have removed all reference to non-basin-aggregated results. Text explaining this has been added to Sect. 2.5.

The following text has also been added to Sect. 2.5, clarifying that while the definitions of "flood risk" and "drought risk" used here are imperfect, they have been used many times previously in the literature:

> As Asadieh and Krakauer (2017) note, these metrics do not translate directly into impacts due to the mitigation capacity and nonlinear effectiveness of reservoirs, flood control mechanisms, and other infrastructure, as well as changes in demand and mean climate. However, changes in streamflow extremes have served as rough indicators in a number of previous global-scale studies (e.g., Tang and Lettenmaier, 2012; Hirabayashi et al., 2013; Dankers et al., 2014; Koirala et al., 2014).

To clarify the proper amount of meaning with which the reader should consider these results (referring the reader back to the new Sect. 2.5 text above), as well as to smooth the transition between results regarding average runoff and extremes, the following text has been moved/added to create a new second paragraph in Sect. 3.2.2 (new text in **bold**):

> Such regional patterns in runoff change are arguably more important than global means, since impacts of low water and flooding are actually felt at the level of individual river basins. **To evaluate regional impacts, we calculated how much land area was subjected to intensified**

> **wet and/or dry extremes (Sect. 2.5). As discussed in Sect. 2.5, these values should not be taken as direct measurements of flooding or drought impacts, but they do serve as useful indicators.**

We have also added brief explanations of meteorological and socioeconomic drought where those terms are introduced.

*The logic of focusing on biodiversity hotspots is different from the logic employed with other ecosystem services, in the sense that with the other services you allow for increases/decreases and for changes in geographic regions where increases/decreases can happen. Wouldn't this be a useful approach here too? Is it possible that there would be increases in biodiversity in some regions that are not currently considered hotspots?*

Yes, it's possible that increasing area of non-agricultural land could lead to a long-term increase in biodiversity in some regions. However, it's not possible to say where biodiversity is currently "limited" by available land—i.e., where, with enough available land, vegetation communities would see sufficient richness of vascular plant species to qualify under the CI definition. Text to this effect has been added to the explanation of our "biodiversity" indicator metric.

*Conclusion. If you split the results and discussion section into two, then you could consider including the conclusions in your discussion section. The current conclusions are not very startling (storylines with high socioeconomic challenges to climate change mitigation consistently have the most severe consequences for ecosystem services) or are simply a repeat of how important this information could be (which was already in the introduction).*

We have opted not to make a separate Discussion section, instead incorporating the additional discussion suggested in comment by Prof. Harrison and others into the Methods or Results. However, in a revised version, we will add some text to the Conclusions about the various elements of uncertainty that need to be explored in future work, including PLUM parameter uncertainty, vegetation and economic model choice, and selection of global climate model. This will allow the Conclusion section to be less repetitive than in the initial version of the manuscript.

**Minor comments**

*Line 15-16. The statement about future population changes is expressed rather badly and is difficult to grasp, please rephrase.*

The clause between the em dashes has been changed to: "with a population increase by 2100 ranging from 1.5 billion to nearly 6 billion people (KC and Lutz, 2017)".

*Line 25. Is this really a feedback sensu stricto?*

Yes: Land-use change and management affect climate via greenhouse gas emissions and biogeophysics, climate change affects agricultural productivity, changing agricultural

productivity affects land use and management, affecting greenhouse gas emissions and biogeophysics, etc. We do not (yet) model this in LandSyMM, but it is indeed a feedback.

*Lines 47-48. The processes operate on the plant functional types rather than among them. Can you rephrase this to describe the model more clearly.*

Here "among" has been changed to "for".

*Line 51. When you say C3 cereals sown in winter and spring, presumably these are considered as two PFTs, so it would be clearer to say "C3 cereals sown in winter, C3 cereals sown in spring ...."*

This change has been made.

*Lines 68-75. It is impossible to judge whether these measures provide reasonable proxies for water availability, freshwater ecosystem condition, or flood risk because there is no information on how runoff is generated in LPJ-GUESS. is runoff simply the difference between P and ET in a gridcell? or is there transfer of surface water between gridcells? is there a contribution from groundwater?*

The runoff paragraph in the "ecosystem services" section has been edited to clarify: "LPJ-GUESS calculates runoff as the amount of water by which soil is oversaturated after precipitation, leaf interception, plant uptake, and evaporation; note that runoff flow is not modeled (e.g., from one gridcell to another)."

*Line 75. If you mean that hydrologic drought is not the same as meteorologic or socioeconomic drought, why not simply say so? This sentence is unnecessary, and begs the question: what is e.g. socioeconomic drought.*

Meteorological and socioeconomic drought are now briefly defined.

*Line 76-81. How does LPJ-GUESS calculate total nitrogen loss? Do you separate out nitrogen loss from natural ecosystems and agricultural systems?*

The following sentence has been added to Sect. 2.5: "This is the combined rate of dissolved nitrogen losses (a function of percolation rate and soil sand fraction) and gaseous losses from denitrification (1% of the soil mineral nitrogen pool per day) and fire."

*Line 82. The linking of climate change and human health here led me to believe that you were going to look at ecosystem services that mitigated the impact of climate change on human health. Apart from the mention that BVOCs affect ozone which in turn can have impacts on health, you don't really go into this in any depth. For example, you don't mention e.g. mineral dust and the role that vegetation plays on mitigating dust emission pace China. Perhaps changing the emphasis here to plant emissions (which have multiple effects, including on climate and on health) would be a better way to introduce this section.*

"Human health" and "ecosystem services" have been swapped at the beginning of the first sentence of this section.

*Line 94-102. I can see why the focus on the hotspots is attractive, but in this modelling framework is would also be possible to make a more general assessment of biodiversity loss and this would also be valuable.*

A more comprehensive evaluation of the biodiversity impacts of land use change is indeed possible in this framework, but since this paper is broadly-focused, we have decided to not do that here. We believe that effort to be more appropriately directed at a paper focused specifically on biodiversity.

*Line 109. The plant name Miscanthus should be in italics.*

This has been corrected throughout the paper.

*Line 115. Is 500 years really sufficient to bring the carbon pools into equilibrium? or is the phrase a realistic starting point mean to imply that they are not necessarily in equilibrium?*

Spinup information is now located in the Supplementary Methods, Sect. SM1:

All runs are preceded by a 500-year spinup period using a temperature-detrended version of the relevant climate forcings (CRU-NCEP v7 CRUp for the calibration run; IPSL-CM5A-MR for the yield-generating and PLUM-forced historical runs.) This includes a routine that analytically solves for equilibrium soil carbon content, bringing carbon pools into equilibrium before the beginning of the actual run.

*Line 125-126. This first sentence should be moved into the description of the model, as confirmation that PLUM works reasonably well. It is not relevant to a description of the modelling protocol.*

This has been taken care of as part of the Sect. 2.3 overhaul.

*Line 127. Given that PLUM has 7 crop types and LPJ-GUESS only four, how do you input PLUM land use into LPJ-GUESS?*

This information can be found in the Supplementary Methods. The overhauled Sect. 2.3 and new flowchart (Fig. 1 in revised text) point interested readers there.

*Line 129. Please can you bring this flowchart and the table into the main text?*

A flowchart is now in the main text.

*Lines 133-135. Please indicate that the details for these sensitivity tests are given in a following section and reference the section here.*

The following has been added to Sect. 2.3: "Details regarding the inputs of these experimental runs can be found in Sect. 2.4 and the Supplementary Methods."

*Line 139. Surely this should be: Viovy, N. 2018. CRUNCEP Version 7 – Atmospheric Forcing Data for the Community Land Model. Research Data Archive at the National Center for Atmospheric Research, Computational and Information Systems Laboratory. http://rda.ucar.edu/datasets/ds314.3/.*

That appears to be a current version of the dataset, but we accessed the data in a different format, from a different server at a different time. We have added a corresponding citation to: Viovy, N.: CRUNCEP Version 7: Atmospheric Forcing Data for the Global Carbon Budget 2016, 2016. A footnote in Supplementary Methods Section SM1 gives the URL and date of access.

*Line 140. Either spell out what these problems are or refer to a paper that does. Maybe Tang et al. (2017)?*

This is now explained in a footnote in Supplementary Methods Section SM1: "The CRU-NCEP algorithm was designed to match CRU TS3.24 monthly precipitation totals, but it produced unrealistically high numbers of wet days—days with precipitation of at least 0.1 mm—in the tropics and boreal regions in the early part of the 20th century."

*Line 147-149. I am having difficulty with this description. You use time-varying allocations of cropland area per gridcell but a static data set of what these crops were. How did you apply this? Simply assuming that the area might change but the crop remains the same? How much uncertainty does this introduce in your results? Unless you address this in a Discussion section (as suggested above) you need to say something here.*

This is clarified in the last sentence of what is now Sect. 2.4: "Historical crop distributions (i.e., given LUH2 cropland area in a gridcell, what fraction was rice, starchy roots, etc.) came from the MIRCA2000 dataset (Portmann et al., 2010) and were held constant throughout the historical period."

*Line 148. Is this still in prep.?*

Yes.

*Line 154-155. This sentence is a bit unclear. The manure N is held constant in the calibration run but varying in the other runs?*

Yes, that's the correct interpretation—probably hard to understand because of a typo. The sentence (now in Sect. SM1) now reads as follows: "Manure N was added in the historical period according to the annually-varying maps given in Zhang et al. (2017b), but in the calibration run was held constant at year 2000 levels to match the use of the AgMIP fertilizer data."

*Line 179-180. Did you estimate these values or are they provided?*

We estimated them.

*Line 184-185. Recycling 30-years of climate is not "constant climate"*

This is true, but we consider "constant climate" to be an acceptable shorthand that should not mislead a reasonably careful reader.

*Line 210. "first two or so ...." please state what period it actually decreases over.*

The relevant part of this sentence has been changed to: "… in SSP5-85 it decreases through about 2050, after which it increases slowly, ending at a slightly lower global extent …"

*Figure 2. This is unreadable at the size reproduced here and, with the grey background, the paler colours are not sufficiently visible. You need to find a way of making the changes more visible. Maybe splitting this into two figures would help. (Note the same comments apply to Figure 4, 5).*

This figure has been updated to use discrete colors (rather than a gradient) with the gray now darker to improve visibility. It has also been enlarged to fill the page (pending editorial approval to exceed the "two-column" width of 12 cm), and rearranging labels has allowed minor additional enlargement.

*Line 222. I confess that I find this agricultural expansion in Alaska a bit implausible, even for a high-end scenario, given the topographic constraints and the issue of permafrost. I would be intrigued to know when the permafrost disappears in this scenario. And how infrastructure (or lack of it) would impact this expansion?*

According to the US Department of Agriculture[1], there are already several hundred farms in Alaska. As the climate warms, permafrost extent is expected to decline across the Northern-Hemisphere boreal zone, especially in RCP 8.5, suggesting that more area might become arable. While the version of LPJ-GUESS used in this study does not have a complete representation of permafrost dynamics, it does include limitations on various plant and soil processes based on air and soil temperature. Thus, LandSyMM might be overly optimistic with regard to the arable area in Alaska by the end of the century, but we do not feel it to be qualitatively implausible. Text to this effect (including citations of two papers projecting permafrost extent) has been added to the first bullet point in Sect. 3.1.

PLUM does not account for limitations on expansion due to lack of infrastructure, implicitly assuming that if conditions are appropriate—in terms of production capacity given demand—for production of agricultural commodities, the necessary infrastructure will follow.

*Line 228. Since South Asia is not a widely-recognised geographical term, it would be helpful to define where exactly you mean here. Are you including southern China here?*

We use "South Asia" to refer to a set of PLUM country groups: India, Sri Lanka, Pakistan, Afghanistan, Bangladesh, Nepal, and Bhutan. Text has been added explaining this.

*Line 234. What climate change produces more favourable growing conditions in South Asia?*

The relevant sentence has been edited to read (changed text in **bold**): "… it also depends markedly on yield boosts due to **increased rainfall (Fig. SR1)** and rising $CO_2$ …"
* * *
[1]https://www.nass.usda.gov/Publications/AgCensus/2017/Full_Report/Volume_1,_Chapter_1_State_Level/Alaska/st02_1_0009_0010.pdf

*Line 236. Even larger ... even larger than what?*

  This clause has been changed to "Sub-Saharan Africa sees crop production increases even larger than South Asia".

*Line 250 et seq. I find this discussion of other model results here confusing. I think you want to separate this from the presentation of all the results from your experiments and move this type of comparison into a separate discussion session. This would allow you to discuss the plausibility of the other assumptions compared to the assumptions encapsulated in your simulations.*

  The advantage of our current structure is that it allows us to immediately address the most striking pattern in our maps of projected land use change, which is the pasture expansion in central Africa. As we explain, there are reasons behind the patterns that we see in our land use trajectories. We want to provide those to the reader here so that readers have the appropriate context for interpreting the rest of our results.

*Line 267. I do not understand what you mean by "friction" here.*

  We have replaced "'friction'" with "cost".

*Line 269-274. Please take out this speculation about the impacts of including forest products in LandSyMM based on work that has not yet been done.*

  We appreciate that the current phrasing is overly speculative regarding work in progress. However, we feel the idea expressed here is important to fully explain the issue at hand. We have thus changed "Work currently underway to include… may" to "Including… could".

*Line 275-283. And so what? You appear to be saying that you have different results from one study because they used unrealistic inputs, and that you have different results from another study because they made a set of different assumptions. In the first case, perhaps you could assume that a reader might guess that your results are "better", although you never actually use the "unrealistic" word. In the second case, however, you might give use a hint about the assumptions made by IMAGE would produce more/less realistic results and why.*

  We acknowledge that most readers will probably recognize why the results of this study are different from and more internally consistent than those of Alexander et al. (2018). However, as the explanation takes only one sentence, we have decided to leave it in.

  We do not consider LandSyMM more or less "realistic" or "plausible" than other state-of-the-art models. It may be that assumptions similar to those made in IMAGE (such as deviation from historical GDP-diet composition relationships) would be necessary in order to restrict PLUM to a solution space that satisfies the radiative forcing values of each RCP scenario; however, LandSyMM does not yet include a climate model, and so we cannot yet assess that possibility. While we do not include forestry or payments for carbon storage, LandSyMM does have other advantages, as explained in the text. We thus consider this work

to be another contribution to the body of research exploring possibilities for the future of land use and terrestrial ecosystems, and leave it to the reader to make their own judgment about relative plausibility if they care to do so.

*Line 290. "intermediate carbon fertilisation ..." Not phrased felicitously, since it implies that C-fertilisation itself has multiple levels of working (off, half-on, on). Please rephrase.*
To us, "intermediate" does not necessarily imply a measurement of discrete values. Rather, it simply implies "somewhere near the middle of two extremes," which allows for our usage in reference to continuous values.

*Line 299. Sorry, I might have missed this – what do you do about the conversion of secondary vegetation to pasture in terms of carbon. Are we looking at gross or net here?*
The following has been added to Sect. 2.2:
> Land use areas are calculated as net change, which neglects certain dynamics—such as shifting cultivation—that can have significant impacts on modeled carbon cycling especially in some regions (Bayer et al., 2017). Other ecosystem services could be affected as well. LandSyMM does not capture these dynamics, but this was considered an acceptable trade-off for computational efficiency.

*Line 302-303. I am not sure why you are picking out one model from this study. I think you should give the range of estimates. I don't know whether the quoted value for IPSL-CM5A-LR is low-end or high-end.*
> The beginning of this paragraph has been modified to clarify (new text in **bold**):
> > Brovkin et al. (2013) examined the change in land carbon storage over 2006–2100 for a number of climate and land surface models. **This included IPSL-CM5A-LR: the same IPSL-CM5A Earth system model that produced our forcings, except run at a lower resolution (hence, -LR instead of our -MR). They found that IPSL-CM5A-LR,** when forced…

*Figure 3. Please don't abbreviate emission on this Figure.*
> In this figure (now Fig. 4), "emissions" is now spelled out.

*Line 305. "probably"? You could establish this by looking at what the difference in loss of non-agricultural land between these simulations and yours is if you take out the pasture expansion and the expansion of cropland in Alaska.*
> "The difference" here refers not to the difference in area, but rather to the difference in C sequestration. We qualify with "probably" because while there are *definite* differences in terms of where and how much non-agricultural land is lost, quantifying the difference in *C sequestration* due to this would require maps of C stocks and fluxes for the Brovkin et al. model outputs.

*Line 308 et seq. So the difference is caused by the differences in the scenarios, right? But later on you imply its because the models don't include nitrogen-limitation. I think you need to make it clear what you think is giving rise to these differences, scenarios or model set-up.*

*It would, of course, make it interesting to run your experiments with the older scenarios –*
*and this would be helpful in terms of uncertainty analysis.*

The text comparing our C sequestration results to those of Brovkin et al. (2013) is intended to convey that the differences could be due to *both* (a) differences in where and how much non-agricultural land is lost, as well as (b) the fact that photosynthesis is limited by N in our model but not those in Brovkin et al. (2013).

We performed the "back-of-the-envelope" calculation suggested by Prof. Harrison below (methodology explained below), which showed that only a small part of the difference can be explained by land-use change scenario. The end of the Brovkin et al. discussion now reads as follows:

> A rough estimate (not shown) shows that running LPJ-GUESS under RCP8.5 with the same land use change as Brovkin et al. (2013) would have increased total carbon gain by 10–15% at most. Instead, most of the difference is likely because none of the models in Brovkin et al. (2013) limit photosynthesis based on nitrogen availability.

*Line 309–310. Your comparisons with other simulations are unbalanced – having described*
*the results from the Brovkin et al study in some detail you here say that the results are low*
*compared to the Nishina et al. (2015) study even when comparing just to the models in that*
*study with nitrogen limitation. But no details. How low? what was the range simulated by the*
*N-enabled models in Nishina et al.? Do you have any idea why you get a different result?*

Upon re-reading Nishina et al. (2015), it was discovered that instead of assuming constant land use (as we first understood it), those simulations did not include land use at all. This explains the large difference between their results and ours, but makes the comparison rather trivial. We have removed the reference.

*Lines 311-318. So, different models produce estimates less than LandSyMM as well as above.*
*What do we learn from this? You imply this is because of differences in scenarios (while*
*hedging your bets in terms of climate forcing), but what is needed here is a back-of-the-*
*envelope calculation of whether the differences in cropland and pasture area would produce*
*a comparable estimate in LandSyMM. If you wanted to be ultra-realistic, you could use the*
*areas where they show biggest changes in area.*

The following text has been added: "A rough estimate (not shown) shows that running LPJ-GUESS under RCP8.5 with the same land use change as Brovkin et al. (2013) would have increased total carbon gain by 10–15% at most." Because we did not save by-LU carbon pools, we estimated this by taking gridcell mean carbon density and dividing it by non-agricultural fraction to get non-agricultural carbon density, effectively making the extreme assumption that agricultural land had zero carbon. (This estimate thus produces an upper limit to the difference that would have occurred using LUH1 land use areas.) We then multiplied that carbon density by the area of non-agricultural land in the PLUM outputs and LUH1 for 2006–2010 and 2096–2100, and calculated the difference. We excluded grid cells where PLUM had <0.1% non-agricultural land.

*Line 319. Photosynthesis scaling parameters ...... what scaling parameters?*

The end of this sentence has been changed to read (new text in **bold**): "… different climate forcings and **a different photosynthetic scaling parameter (which accounts for real-world reductions in light use efficiency; Haxeltine and Prentice, 1996)**."

*Line 338. Why did you not run it in coupled mode then?*

Our group does not have a version of LPJ-GUESS coupled to a climate model.

*Line 341. Please can you explain what was done in the Asadieh and Krakauer (2017) analyses. Were these full hydrological models?*

We have added text explaining that Asadieh and Krakauer (2017) included full hydrological models.

*Since this was an ensemble, presumably there is a range of estimates for at-risk of flood and at-risk of drought? Please give these ranges in the text*

Asadieh and Krakauer (2017) only presented their multi-model average results, not the range of results across all models.

*How much of a difference does using monthly versus daily values make to the estimates of area affected?*

We have added a sentence to Sect. 3.2.2: "We expect that our results for land area with increasing and decreasing flood risk would have been lower and higher, respectively, had we used daily values for P95 as Asadieh & Krakauer (2017) did, instead of the LPJ-GUESS-output monthly values." Quantifying this difference does not seem possible without adding code to LPJ-GUESS allowing daily runoff outputs, then performing the runs again.

*Line 349. How many classes are there? You need to spell out that you are talking about increases/decreases in flood risk and drought risk areas).*

There are four classes, as given in what is now Table 2. A reference to Table 2 has been added to the sentence in question.

*Line 379. Please explain why high CO2 suppresses BVOC formation and provide references here.*

The following has been added here: "The exact cellular regulatory processes of this '[$CO_2$] inhibition' remain enigmatic; recent evidence suggests that reduced supply of photosynthetic energy and reductants plays a major role (Rasulov et al., 2016)."

*Line 382-383. This sentence is confusing because it seems to say that boreal forests are causing declining monoterpene emissions – whereas I think the idea is that decreases in boreal forest area coupled with less effective BVOC production in the surviving boreal forest area are responsible. Please rewrite.*

We have replaced "drivers" with "areas".

*Line 393. Please italicise Miscanthus*

This has been corrected throughout the manuscript.

*Line 396 et seq. This paragraph states that predicting the effects of changing BVOCs is difficult because the model framework doesn't include atmospheric chemistry. I am not sure what "a surface-level discussion of possible effects" means here. You can predict potential changes in BVOC emissions, and so perhaps this is the point to stop at. There is no need to go further and speculate about what the impact of these changes might be on atmospheric chemistry, climate and/or health.*

We think it is helpful for readers to be given some sort of context for the results, but acknowledge that this discussion is indeed speculative nature of this discussion. To make this clear, we have:

- Swapped the last two paragraphs of this section.
- Added the following to the end of what is now the second-to-last paragraph: "However, we wish to provide context for the benefits and detriments associated with changing BVOC emissions, as well as some limitations related to our model setup."
- Replaced the last sentence of what is now the last paragraph with the following (new text in **bold**): "Moreover, the loss of natural land is itself associated with myriad negative health impacts (Myers et al., 2013) **which are not simulated in LandSyMM**, so it would be shortsighted to view deforestation-induced BVOC reductions as a public health boon. **Testing whether and to what extent any of the mechanisms described in this paragraph would make a difference to regional climate and human health would require significant extension of LandSyMM, including the incorporation of new sub-models.**"

*Line 403. The fact that this one region is not defined as a hotspot, and that it has a big impact on the results, makes a good case for extending this analysis to consider changes in biodiversity everywhere.*

Theoretically it would be possible to extend this analysis to areas not currently classified as hotspots by surveying the literature to determine the floristic diversity of all ecoregions and then—where the "at least 1500 vascular plant species" requirement is met—including all ecoregions that will have lost at least 70% of their natural vegetation by the end of the century. However, the effort that would require would better be spent on a more comprehensive analysis of not only area loss but corresponding extinctions (see below). Such an analysis would be valuable, but is outside the scope of this paper.

*Line 410-415. Please rewrite this section to first make the comparison and then explain possible sources of differences. ... Line 415. Are you saying that species-area curves are an inappropriate tool for estimating extinction rates rather than species numbers?*

The text has been edited to clarify that species-area curves are correct in accounting for how the number of species lost per hectare of land conversion decreases as total area converted increases. This comparison has also been rearranged to highlight the reason for citing Jantz et al. (2015) at all: to illustrate (a) the importance of this nonlinear relationship

and (b) that our analysis did not take this into account, meaning that our results do not correspond directly to extinction estimates.

*Lines 419-420. I hadn't realised that climate changes and CO2 could have an effect on models too! Please rephrase this.*

Hof et al. (2018) used a tool, species distribution modelling, that has not yet been mentioned in this manuscript. Such models use climate and land-use change as inputs.

*Line 423. "We may see a similar effect ...." Please clarify: do you or don't you?*

The following has replaced the part of this paragraph beginning "We may see…":

> We see a similar effect: If ignoring *Miscanthus* area, loss of natural land in CI+CSLF hotspots is reduced (respectively for SSP1-45, SSP3-60, SSP4-60, and SSP5-85) by about 100%, 45%, 39%, and 17%. However, because land cleared for biofuel is not available for other crops, a full accounting of the contribution of biofuel expansion to land conversion and thus biodiversity would require PLUM runs with no biofuel demand.

We have decided not to perform those extra PLUM runs, believing that effort would serve better in work more focused on the future impacts of land-use change on biodiversity rather than the more general review here.

*Line 429-430. Considering that the results from LandSyMM have been compared to a range of other model simulations in this paper, the first sentence really doesn't make sense. Maybe this is more comprehensive in terms of the range of scenarios and the range of outputs, but what else is different here?*

This first sentence is indeed intended to highlight this work's novelty due to its comprehensiveness; "comprehensively" has been added to stress this point. The beginning of the second sentence has been modified to highlight other advantages of LandSyMM as mentioned in the revised Introduction (new text in **bold**): "**Using a uniquely spatially-detailed, process-based coupled model system,** we show…"

---

## Author Response (AR2)

**Response to minor corrections 2020-03-23**

In addition to the changes described below, we have replaced "FILL ON ACCEPTANCE" with "44" in the Acknowledgements section.

*1) abstract: the sentence while indicators of some ecosystem services show trends .. and others not: This is very general. Can you be more precise?*

This text has been edited to read as follows: "Some ecosystem services depend critically on land use and management: for example, carbon storage, the gain in which is more than 2.5 times higher in a low-LUC scenario (SSP4-60) than a high-LUC one with the same carbon dioxide and climate trajectory (SSP3-60). Other trends are mostly dominated by the direct effects of climate change and carbon dioxide increase. For example, in those two scenarios, extreme high monthly runoff increases across 54% and 53% of land, respectively, with a mean increase of 23% in both."

*2) you use many times indicators see. Or scenarsios saw etc. Scenarios can not see, it is that the scenario simutations show. I would prefer to change this (eg.L7, L380,L381, L399,L488)*

This has been changed throughout the manuscript. In the process, we noticed that the first paragraph in Sect. 3.2.3 mischaracterized the results for SSP4-60 as "not show[ing] a notable change." The first sentence of this paragraph has been kept and the rest changed to read as follows: "SSP3-60 and SSP5-85 show large increases in N loss of 28% and 22%, respectively. N loss increases about half as much in SSP4-60 (11%) and only slightly in SSP1-45 (2%)."

*3) Legend Figure 2: change 2001-2010 mean to the average over 2001-2010*

This change has been made.

*[]4) Legend Figure 2 and Figure 4: high interannual variability. I would say high intra- and interannual variability.*

"Interannual" has been changed to the more general "temporal" in both instances.

*5) Line 273: that region net loses large (dont understand the sentence)*

The sentence now reads as follows (edited text in **bold**): "Although crop demand in South Asia (here, India, Sri Lanka, Pakistan, Afghanistan, Bangladesh, Nepal, and Bhutan) increases by more than 100% in SSP5-85 and 170% in SSP3-60 (Fig. SR9), **after harmonization the cropland area in that region is greatly reduced**: ca. 30% and 20%, respectively."

*6) Figure 5: I wouldnt add for every SSP the value at 2000 as those are equal. I like the way how you have done this for figure 7 in which you show how much it has changed compared to 2000. So for SSP1-45 (+43 PgC +/- 2, 11.7%), for SSP3-60 (+22 PgC , 5.8%) etc.*

This change has been made. In addition, reflecting revisions made to the bar graph figures, the interannual variability has been removed from the labels. The last sentence of the caption has been removed; it had read: "
[revised manuscript text omitted]